# Biosynthesizing structurally diverse diols via a general route combining oxidative and reductive formations of OH-groups

Yongfei Liu[1], Wei Wang[1] & An-Ping Zeng [1,2✉]

Diols encompass important bulk and fine chemicals for the chemical, pharmaceutical and cosmetic industries. During the past decades, biological production of C3-C5 diols from renewable feedstocks has received great interest. Here, we elaborate a general principle for effectively synthesizing structurally diverse diols by expanding amino acid metabolism. Specifically, we propose to combine oxidative and reductive formations of hydroxyl groups from amino acids in a thermodynamically favorable order of four reactions catalyzed by amino acid hydroxylase, L-amino acid deaminase, α-keto acid decarboxylase and aldehyde reductase consecutively. The oxidative formation of hydroxyl group from an alkyl group is energetically more attractive than the reductive pathway, which is exclusively used in the synthetic pathways of diols reported so far. We demonstrate this general route for microbial production of branched-chain diols in *E. coli*. Ten C3-C5 diols are synthesized. Six of them, namely isopentyldiol (IPDO), 2-methyl-1,3-butanediol (2-M-1,3-BDO), 2-methyl-1,4-butane-diol (2-M-1,4-BDO), 2-methyl-1,3-propanediol (MPO), 2-ethyl-1,3-propanediol (2-E-1,3-PDO), 1,4-pentanediol (1,4-PTD), have not been biologically synthesized before. This work opens up opportunities for synthesizing structurally diverse diols and triols, especially by genome mining, rational design or directed evolution of proper enzymes.

[1] Hamburg University of Technology, Institute of Bioprocess and Biosystems Engineering, Denickestrasse 15, Hamburg 21073, Germany. [2] Center of Synthetic Biology and Integrated Bioengineering, School of Engineering, Westlake University, Hangzhou, Zhejiang 310024, China. ✉email: zenganping@westlake.edu.cn

D iols are important bulk and fine chemicals that have wide applications as solvents, monomers for polymer synthesis, feed additives, and ingredients of cosmetics and pharmaceuticals[1–3]. There have been large interests and efforts in biotechnologically producing diols in the last decades[1,2,4]. In fact, the successful development and industrial application of microbial production of diols such as 1,3-propanediol (1,3-PDO), 1,4-butanediol (1,4-BDO), and 1,3-butanediol (1,3-BDO) from renewable carbohydrates represent some of the most important achievements of industrial biotechnology in recent years[2,3,5–7]. Advances in synthetic biology and metabolic engineering enable the development of new biological routes for either existing microbial diols or even completely new diols, including unnatural pathways for the use of one-carbon resources such as $CO_2$, formic acid, formaldehyde, and methanol[8,9].

However, most of the biological routes studied so far dealt exclusively with the formation of linear-chain 1,n-alkanediol. In fact, branched-chain alkanediols are also of great industrial interest. For example, 3-methyl-1,3-butanediol (also known as isopentyldiol, IPDO) is an appealing compound widely used in chemical and cosmetic industries as a moisturizer, solubilize, and preservative booster. It is presently synthesized chemically from fossil resources. There is a great interest to generate bio-based IPDO for better safety and performance in terms of skin feeling, deodorization, and antibacterial properties. There is no natural biological route to generate IPDO. Similar situations apply also to other branched-chain 1,n-alkanediols such as 2-methyl-1,3-propanediol (MPO), 2-ethyl-1,3-propanediol (2-E-1,3-PDO), and 1,4-pentanediol (1,4-PTD). Furthermore, existing metabolic pathways for diols suffer from the drawbacks of relatively high energy requirement in terms of ATP or reducing power NAD(P)H and often limited selectivity, leading to challenges in host strain engineering for high diol yield and productivity.

A common feature of diols is that they possess two hydroxyl groups. Concerning the formation of hydroxyl group, it can be principally formed either through oxidation of an alkyl group (referred to as oxidation pathway) or through reduction of a carboxyl group or a carbonyl group (referred to as reduction pathway) (Fig. 1a). However, thermodynamic barriers are relatively high for a direct formation of a hydroxyl group in both cases. Since the reductive formation of hydroxyl group is normally associated with decarboxylation, it should be thus thermodynamically more favorable. This occurs however with the loss of a carbon atom that leads to a lower carbon yield. More importantly, the intermediates formed often contain carbonyl group (e.g., in form of aldehyde) which are very reactive under physiological conditions and cannot accumulate to a high level due to toxicity to cells. To our knowledge, synthetic biological routes proposed for diol production almost exclusively use one or two reductive reactions for the formation of the hydroxyl group(s) (thus the reduction pathway) and they overcome the reduction potential barrier by utilizing an extra ATP in addition to NAD(P)H (Fig. 1a). For instance, one major development in recent years is the microbial production of 1,4-BDO via succinate[7]. In this pathway, the two hydroxyl groups are obtained through activation of the two carboxyl groups using CoA-transferase, followed by two sequential reduction steps catalyzed by NAD(P)H-dependent reductases. This pathway requires 1 molecular ATP in addition to 4 molecular NAD(P)H to generate 1 molecular 1,4-BDO from succinate. A similar strategy was used in a synthetic pathway for the biosynthesis of 1,3-BDO[10]. Recently, Wang et al. proposed an interesting platform pathway to synthesize 1,3-PDO, 1,4-BDO, and 1,5-pentanediol (1,5-PDO) from charged amino acids, in which one hydroxyl group is obtained by the reduction of a semialdehyde group with the input of 1 molecular NADPH, and the other is derived from the

reduction of a carboxyl group successively catalyzed by a promiscuous carboxylic acid reductase (CAR) and an endogenous aldehyde reductase at the expense of 1 molecular ATP and 2 molecular NADPH[5]. In this pathway, the total energy driving force of the reaction cascade for the conversion of the carbonyl group to the corresponding hydroxyl group is quite low ($\Delta_r G$': $-5 \sim -1$ kcal/mol), rendering the cascade reactions thermodynamically constrained. Because semialdehyde and aldehyde cannot accumulate to a high-level intracellularly due to their high toxicity, the metabolic flux could be also limited from physiological and kinetic considerations.

In this study, we design a diol biosynthetic platform based on a biosynthetic principle of combining an oxidation pathway and a reduction pathway for the respective formation of the two hydroxyl groups of target diols (Fig. 1b). The feasibility of the proposed strategy is verified by the biosynthesis of 10 C3–C5 diols from glucose in Escherichia coli. Moreover, IPDO is taken as an example to demonstrate the major advantages and to illustrate the key issues for exploiting the potential of the diol biosynthetic route.

## Results

**Design and verification of the platform pathway for biosynthesis of various diols**. The proposed diol biosynthetic platform proposed in this study makes use of the highly active amino-acid synthetic pathways and channels the metabolic flux into diol formation through four reaction steps catalyzed by amino acid hydroxylase, L-amino acid deaminase, α-keto acid decarboxylase, and aldehyde reductase, respectively. In such a reaction cascade, one of the hydroxyl groups of the product diol is first formed through oxidative hydroxylation of an intermediate amino acid that is coupled with the oxidation of α-ketoglutarate (KG) to succinate without the requirement of an extra NAD(P)H, which is highly exergonic ($\Delta_r G$': $-100 \sim -120$ kcal/mol). The other hydroxyl group is formed from successive deamination, decarboxylation, and reduction of the resulting intermediates (Fig. 1c). Both the deamination and decarboxylation are thermodynamically highly favorable chemical reactions that can give the necessary push for the thermodynamically less favorable reductive formation of hydroxyl group in the last step.

In addition to the above principal advantages of the general concept, the synthetic route has several specific advantages and attractiveness. First, it stretches from the well-elucidated natural amino-acid biosynthesis pathways. The highly active metabolism of amino acids has been used to biotechnologically produce a number of valuable amino acids such as L-lysine and L-glutamate at a large industrial scale (millions of tons per year). The existing experiences and know-how can be exploited for further pathway engineering aimed at producing corresponding diols. In fact, amino acids with carboxyl and amino groups in their structure have been successfully used as intermediates for the synthesis of other valuable compounds. For example, deamination or decarboxylation of amino acids can generate various appealing carboxylates or diamines such as γ-amino-butyrate (GABA), 1,5-diaminopentane and 1,4-diaminobutane, trans-p-hydroxycinnamate, and urocanate[11,12]. Second, the availability of different stereoselective hydroxylases enables the hydroxylation of an amino acid at a certain carbon position can be realized by selecting or designing a proper hydroxylase, offering thereby the possibility to generate structurally diverse diols from the same amino acid (Fig. 1c)[13,14]. Last but not least, relatively high purity of target product can be obtained owning to the stringent stereocontrol of hydroxylation positions by hydroxylases, which will greatly reduce the difficulty and cost of the subsequent separation and purification process.

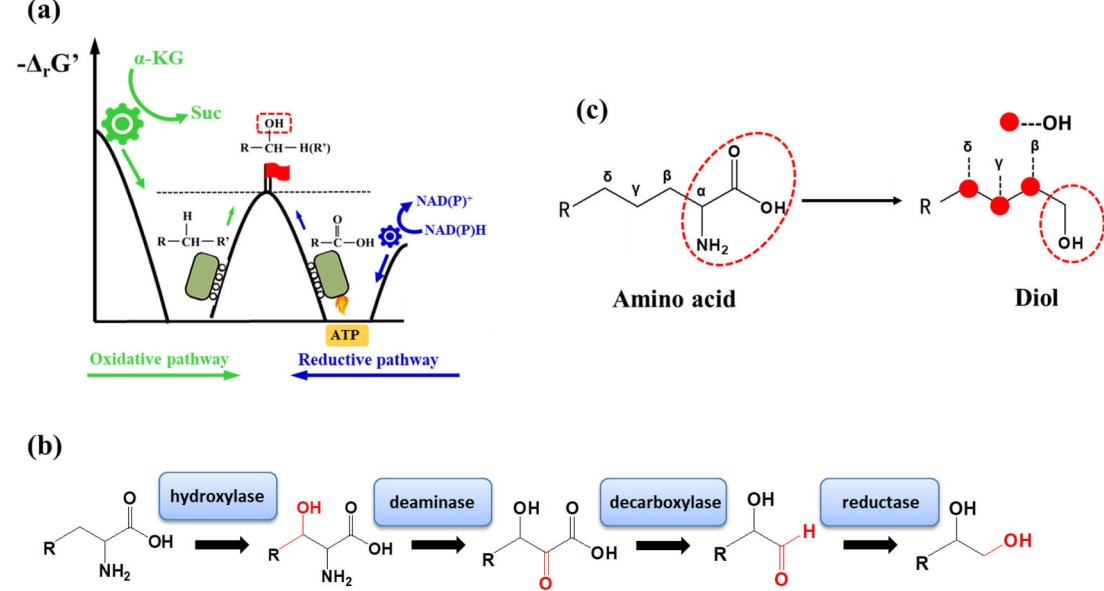

**Fig. 1 Schematic representation of the principle and platform pathway proposed in this study for producing diols from amino acids. a** Principle of two strategies that natural enzymes employ to overcome the thermodynamic barriers (black dashed line) for the formation of hydroxyl group from hydrocarbon by oxidation (left) and from carboxyl group by reduction (right). In the oxidative pathway, the oxidation of alkyl group is often coupled with an exergonic reaction (shown is the oxidation of α-ketoglutarate (α-KG) to succinate used in this study) (green arrow). In the reductive pathway using NAD(P)H as reducing power, ATP is normally needed as an additional energy supply (blue arrow). **b** The proposed pathway comprises four cascade reactions, which are successively catalyzed by amino-acid hydroxylase, ʟ-amino acid deaminase, α-keto acid decarboxylase, and aldehyde reductase, respectively. Functional groups colored in red represent the groups modified in each step. Hydroxylation at β-position is shown only for illustration purposes. **c** Illustration of synthesis of diverse branched-chain diols. One hydroxyl group is formed from the amino carboxyl group via Ehrlich pathway (red dotted circles), while the other one is attached to the β-, γ-, or δ-C atom of alpha-amino acid as determined by the property of hydroxylase used.

As depicted in Fig. 1b, the first key reaction of the proposed diol synthetic route is the hydroxylation of an amino acid to form a corresponding hydroxyl amino acid in the presence of an amino acid hydroxylase. The hydroxyl amino acid is then converted to the corresponding hydroxyl α-keto acid by an ʟ-amino acid deaminase, followed by decarboxylation and reduction by an α-keto decarboxylase and an aldehyde reductase, respectively, to generate the desired diol.

So far a large set of canonical amino-acid hydroxylases have been discovered by genome mining or rational engineering[15–20]. A typical canonical amino-acid hydroxylase requires α-KG and molecular oxygen as the two common co-substrates, and one amino acid as the specific substrate[21,22]. Interestingly, and very useful for diol production, amino acid hydroxylases of different species can catalyze hydroxylation of the same substrate but at different carbon positions with high stereoselectivity[23]. We selected hydroxylases from various species to generate 10 different diols from 6 amino acids in E. coli. A summary of the substrate amino acids and their corresponding diol products is given in Fig. 2a. The reaction schemes of the synthesis of the individual diols through this proposed pathway are shown in Supplementary Figures 1 and 2. The plasmids and primers used in this study are listed in Supplementary Data 1 and Supplementary Data 2, respectively. Specifically, the hydroxylase MFL from *Methylobacillus flagellatus* KT, which catalyzes the hydroxylation of branched-chain amino acids (BCAAs) at the C-4 position, was expressed in the pET28a plasmid, while ʟ-amino acid deaminase AAD$_{vul}$ from *Proteus vulgaris*, α-keto acid decarboxylase KDC from *Lactobacillus lactis,* and aldehyde reductase YqhD from *E. coli* W3110 were co-expressed in the plasmid pZA. The two plasmids were co-transformed into *E. coli* BL21(DE3), yielding strain DL01. DL01 was cultivated in FM-II medium added with different BCAAs as substrate. When 25 mM

valine, leucine, norvaline or norleucine were used as a substrate, respectively, 18.5 mg/L MPO, 230.5 mg/L IPDO, 8.7 mg/L 1,3-BDO and 4.2 mg/L 1,3-pentanediol (1,3-PTD) were produced after 48 h of cultivation respectively (Fig. 2b, Supplementary Figure 3 and Supplementary Figure 4–7), demonstrating that this platform pathway functions in vivo. Different from the above BCAAs, isoleucine possesses two C-4 positions (C4$^1$-OH and C4$^2$-OH). It has been reported that MFL has no catalytic activity on isoleucine[17]; but fortunately, selective hydroxylation of the two C-4 atoms can be realized separately by using two different hydroxylases encoded by *hilA* and *hilB* from *Pantoea ananatis* AJ13355[15]. Therefore, we constructed strains DL02 and DL03 that separately overexpressed *hilA* and *hilB*, in addition to AAD$_{vul}$, KDC, and YqhD in *E. coli* BL21(DE3). Cultivations of DL02 and DL03 in FM-II medium supplemented with 25 mM isoleucine led to the formation of 145.2 mg/L 2-E-1,3-PDO and 12.1 mg/L 2-methy-1,3-butanediol (2-M-1,3-BDO), respectively (Fig. 2c, Supplementary Figure 3, and Supplementary Figure 8 and 9). The significant difference in the production titer of 2-E-1,3-PDO and 2-M-1,3-BDO indicates that HilA has a higher catalytic capability on isoleucine than HilB. This can be attributed to the fact that isoleucine is the natural substrate of HilA, while the natural substrate of HilB is 4$^1$OH-isoleucine, the product of isoleucine catalyzed by HilA[15].

Hydroxylase GriE from *Streptomyces* DSM 40835 was reported to selectively catalyze the hydroxylation of amino acids at C-5 position[24]. Therefore, we constructed the strain DL04 by substituting MFL with GriE. After 48 h of fermentation, DL04 was able to produce 276.4 mg/L 2-methyl-1,4-butanediol (2-M-1,4-BDO), 7.8 mg/L 1,4-BDO, and 11.9 mg/L 1,4-PTD in FM-II medium supplemented with 25 mM leucine, norvaline, and norleucine, respectively (Fig. 2d, Supplementary Figure 3, and Supplementary Figure 10–12). Remarkably, the reported catalytic efficiency of GriE

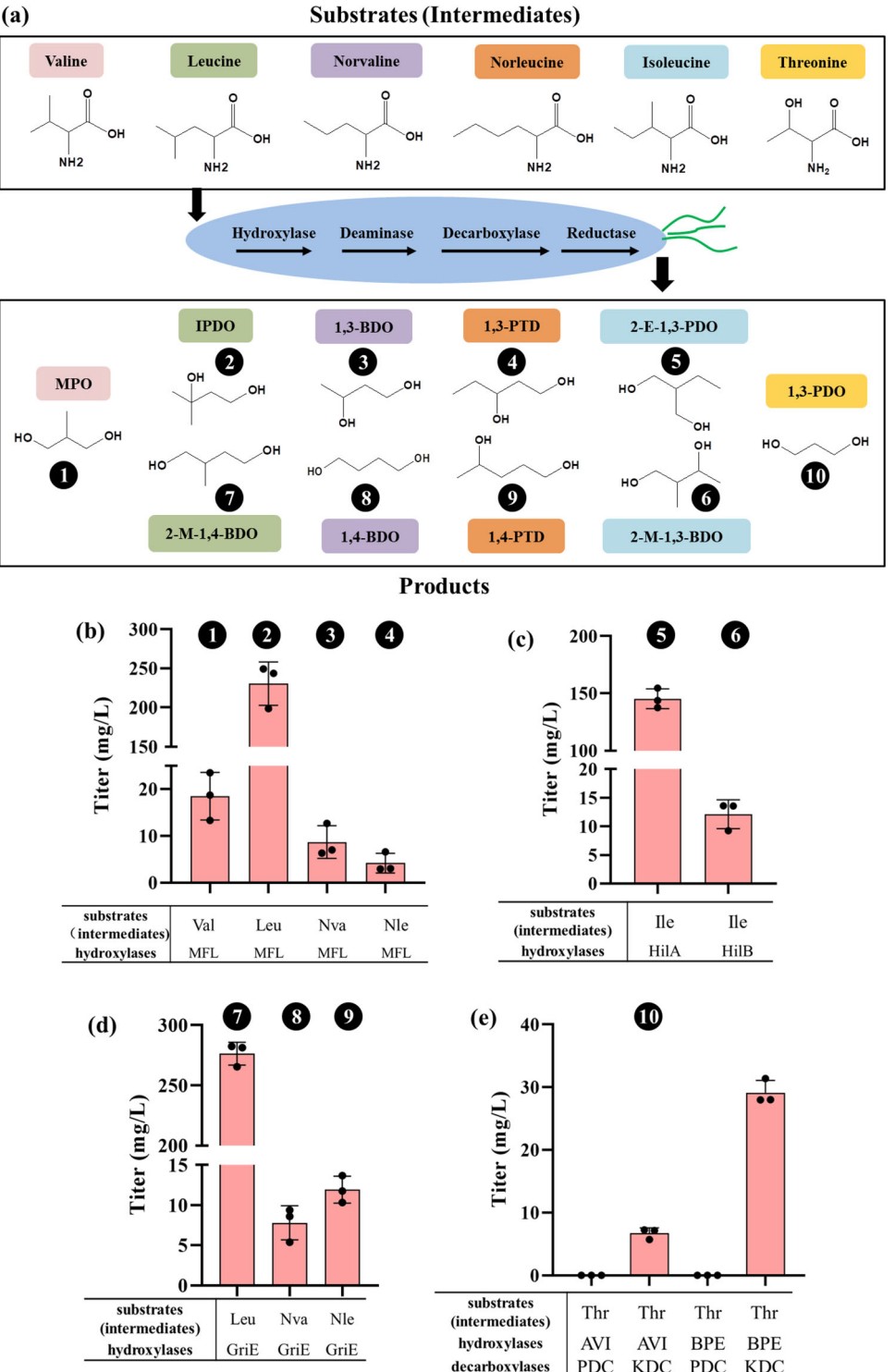

**Fig. 2 Details of the platform pathway for the production of 10 structurally diverse diols.** The substrate (intermediate) amino acids used and their corresponding diol products are highlighted by the same colors (**a**); **b–e** production titers of diols. The numbers representing each diol are in line with those in **a**. *MPO* 2-methyl-1,3-propanediol, *IPDO* isopentyldiol, *1,3-BDO* 1,3-butanediol, *1,3-PTD* 1,3-pentanediol, *2-E-1,3-PDO* 2-ethyl-1,3-propanediol, *2-M-1,3-BDO* 2-Methyl-1,3-butanediol *2-M-1,4-BDO* 2-Methyl-1,4-butanediol, *1,4-BDO* 1,4-butanediol, *1,4-PTD* 1,4-pentanediol, *1,3-PDO* 1,3-propanediol, *Val* valine, *Leu* leucine, *Ile* isoleucine. *Nva* norvaline, *Nle* norleucine, *Thr* threonine. **b–e** the average and standard deviation (s.d.) of three biologically independent experiments are shown. Source data are provided as a Source Data file.

towards the three BCAAs (leucine>>norleucine>norvaline)[24] was in accordance with the titers of the corresponding diols (2-M-1,4-BDO>>1,4-PTD>1,4-BDO), indicating that the catalytic efficiency of hydroxylase is a determinant factor for the achievable titers of the diol products.

From the above results, it is confirmed that the hydroxylation of BCAAs by MFL and GriE is under stringent stereocontrol that enables the synthesis of structurally different diols from the same BCAA, as in the cases of leucine, norvaline, and norleucine (Fig. 2b, d). This clearly demonstrates the power and advantages of the biological production of structurally diverse diols using the strategy proposed in this work. To our knowledge, microbial synthesis of the six branched-chain diols MPO, IPDO, 2-M-1,3-BDO, 2-M-1,4-BDO, 2-E-1,3-PDO, and 1,4-PTD has not been reported before. Most of these products belong to important commodity chemicals with a variety of applications. For example, MPO is widely used to produce polyesters, resins, and lubricants[25], IPDO serves as an excellent moisturizer in the cosmetic industry, and 1,4-PTD is reported as a valuable building block for the synthesis of chloroquine, one of the most effective medicines for the prevention of malaria[26].

1,3-PDO is probably the most prominent diol whose bioproduction has attracted great attention for decades, e.g., microbial production of 1,3-PDO from biodiesel production-derived glycerol or glucose. However, there are no natural microorganisms that can produce 1,3-PDO directly from sugar. In addition to the well-known synthetic pathway of 1,3-PDO from glucose over glycerol as industrially used by DuPont and Tate&Lyle, further de novo pathways have been developed to synthesize 1,3-PDO, e.g., by expanding the homoserine synthesis pathway using glucose as substrate[9,27] or by developing an aldolase-catalyzed one-carbon assimilation pathway using glucose and methanol as co-substrates[8]. In this study, we sought to produce 1,3-PDO by extending the threonine catabolism in E. coli. This pathway shares a similar structure with the proposed general route except that the deaminase AAD$_{vul}$ was replaced by the endogenous threonine deaminase IlvA, because IlvA can also remove the −OH group at C-3 position in accompany with the deamination, and the hydroxylase MFL, which is inactive towards threonine, was replaced by one of another two hydroxylases, namely AVI from *Agrobacterium vitis* and BPE from *Bordetella petrii,* which possess hydroxylase activities towards threonine[17].

Since the conversion of threonine to 4-hydroxy-threonine catalyzed by threonine hydroxylase and the conversion of 4-hydroxy-α-ketobutyric acid to 1,3-PDO catalyzed by α-keto acid decarboxylase and aldehyde reductase have been demonstrated by the previous studies[8,17,28,29], we carried out an in vitro examination of the catalytic activity of threonine deaminase on 4-hydroxy-threonine and confirmed that 4-hydroxy-threonine produced from the hydroxylation of threonine is further converted by threonine deaminase (Supplementary Figure 14). Next, to verify this pathway in vivo, these two hydroxylases AVI and BPE were assembled into the platform pathway in alternative combination with two α-keto acid decarboxylases, namely KDC from *L. lactis* and PDC from *Zymomonas mobilis*, which were employed to produce 1,3-PDO in the previous reports[8,9]. The obtained four recombinant strains were cultivated in FM-II medium supplemented with 25 mM threonine. Surprisingly, none of the four strains produced 1,3-PDO. A possible interpretation of this pitfall is that the deaminase IlvA directly competes with the hydroxylase AVI or BPE in the use of threonine as substrate, but the $K_{cat}$ of IlvA is much higher (~16,000-fold) than that of AVI or BPE[17,30], resulting in the observation that the majority of threonine was quickly deaminized by IlvA without being first hydroxylated by AVI or BPE. To overcome the mismatched catalytic efficiencies of the enzymes, we overexpressed the

threonine hydroxylase (AVI or BPE), as well as the α-keto acid decarboxylase (PDC or KDC) and the aldehyde reductase YqhD in E. coli BL21(DE3) so that 4-hydroxy-threonine should be first formed by hydroxylation before the endogenous IlvA fulfills the task of deamination, followed by enhanced decarboxylation and reduction to form 1,3-PDO. As the result, the recombinant strains DL06 and DL08 (both harboring KDC) yielded 6.7 mg/L and 29.1 mg/L 1,3-PDO in shake flask culture, respectively (Fig. 2e, Supplementary Figure 3 and 13). In contrast, another two strains DL05 and DL07 harboring PDC still did not produce 1,3-PDO due to unknown reasons, though both KDC and PDC have been reported to catalyze the conversion of 4-hydroxy-2-ketobutyrate to 3-hydroxy-propionaldehyde[8,9]. One possible interpretation is that the activity of PDC is inhibited by certain metabolites produced by the promiscuous catalysis of hydroxylase in DL05 and DL07. Compared with other 1,3-PDO synthetic pathways, the hydroxylation, deamination, and decarboxylation in the proposed pathway are all thermodynamically more favorable reactions (Supplementary Figure 15), which can give the necessary push for the thermodynamically less-favorable reductive formation of hydroxyl group in the last step.

### Examination of the reaction orders for IPDO biosynthesis. The results of 1,3-PDO synthesis with different combinations of enzymes presented above indicate that the reaction order (e.g., first hydroxylation or deamination) of the proposed diol bio-synthesis pathway may affect the biosynthetic efficiency. In the following, we took IPDO as a further example to address this question in more detail.

Since the reduction of aldehyde by aldehyde reductase always follows the decarboxylation of α-keto acid by α-keto acid decarboxylase, we considered these two reactions as one single step. In this way, a total of six reaction orders (routes) are thus possible, as listed in Fig. 3a. Routes 1 and 2 (R1 and R2) have both hydroxylations as the first reaction step, but differ from each other in the order of deamination and decarboxylation/reduction. To examine whether R1 or R2 or both work, we overexpressed AAD$_{vul}$, KDC, and YqhD in the plasmid pZA to yield pZA-aky and transformed it into E. coli BL21(DE3). The obtained strain DL01-R12 was cultivated in FM-II medium with the addition of 5 mM 4OH-Leu, the hydroxylation product of leucine. This strain produced 374.1 mg/L IPDO after 48 h of cultivation (Fig. 3b). Further investigation by enzyme activity assay, however, showed that in the absence of the deaminase AAD$_{vul}$, KDC, and YqhD had no catalytic activity on 4OH-Leu (Supplementary Figure 16a, b) and thereby ruled out the possibility of Route 2.

In a similar way, we examined Route 3 and Route 4, which have deamination as the first reaction step and differ from each other in the order of hydroxylation and decarboxylation/reduction. We overexpressed only KDC and YqhD in the plasmid pZA to yield pZA-ky and transformed it together with pET-mfl into E. coli BL21(DE3). The resulted strain DL01-R34 was able to produce 19.2 mg/L IPDO (Fig. 3b) in FM-II medium supplemented with 5 mM 2-ketoisocaproate (KIC), the deamination product of leucine catalyzed by AAD$_{vul}$. Moreover, the feasibility of Route 3 was further verified by the fact that MFL showed activity on KIC (see below in Fig. 5c). The possibility of Route 4 was eliminated due to the missing hydroxylation activity of MFL towards isopentanol (IPT, the decarboxylation/reduction product of KIC), as can be observed that no IPDO was produced by the strain DL01-R4 that carried pET-mfl in E. coli BL21(DE3) in FM-II medium supplied with 5 mM isopentanol (Fig. 3b). Finally, we ruled out the possibility of Routes 5 and 6, which have in common the decarboxylation/reduction as the potential first step of the reaction cascade, because both KDC and YqhD had no activity on leucine (Supplementary Figure 16a, c).

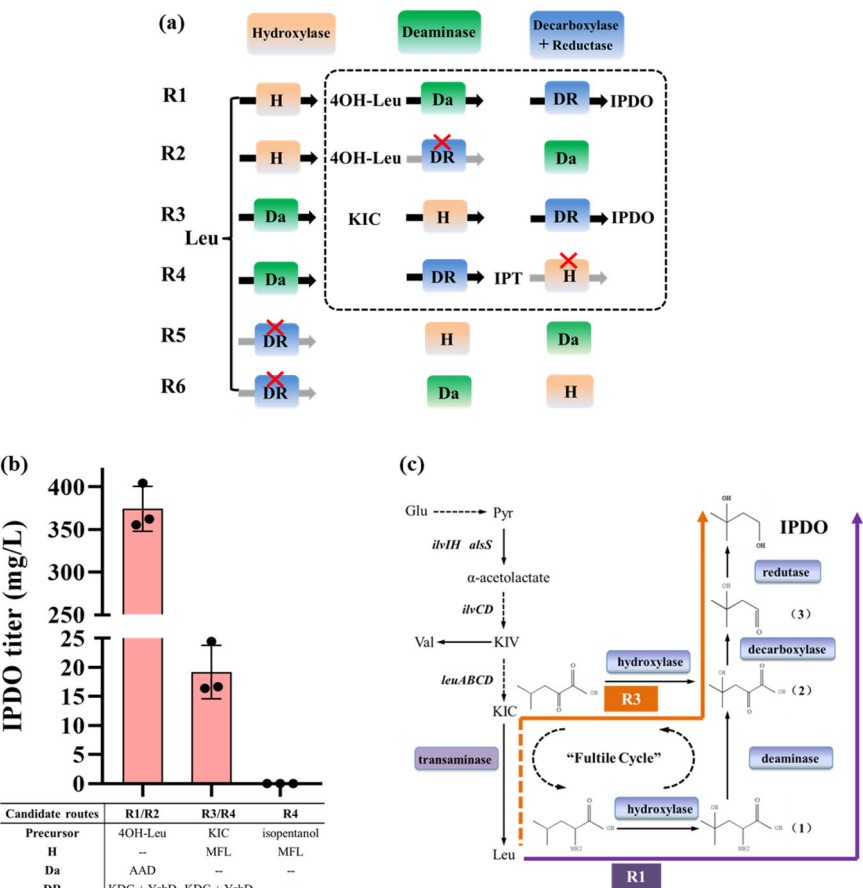

**Fig. 3 Determination of reaction orders to construct a pathway for IPDO production from leucine or its precursor 2-ketoisocaproate (KIC) in *E. coli*. a** Six possible reaction orders (R1-R6) in the proposed pathway for IPDO production. Since the reduction of aldehyde by aldehyde reductase is always after decarboxylation of α-keto acid by α-keto acid decarboxylase, these two reactions are considered as one single step (DR). Black arrows represent the reactions with proved functions in vivo, while gray arrows with red × represent the reactions that do not work in vivo. Precursors and enzymes in dotted boxes were employed for verification of each possible reaction order. **b** IPDO titers obtained from *E. coli* strains to carry different candidate routes. IPDO production was initiated by different strains cultivated in FM-II medium supplementing 5 mM precursor (4OH-Leu for strain DL01-R12, KIC for strain DL01-R34, and isopentanol for strain DL01-R4). 4OH-Leu: 4-hydroxy-leucine, KIC: 2-ketoisocapraote, IPT, isopentanol. **c** Two verified routes for the production of IPDO in *E. coli*. Route R1 was shown in purple and R3 in orange. A transaminase-deaminase futile cycle existed in route R1. The deamination reaction of Route 3 is a fictitious reaction that is indicated by a dashed line. For b, the average and s.d. of three biologically independent experiments are shown. Source data are provided as a Source Data file.

Based on the above results, we concluded that there are two pathways working simultaneously for IPDO production from leucine (Fig. 3c), i.e., the reaction steps were performed either in the order of hydroxylation, deamination, decarboxylation, and reduction from leucine (Route 1) or in the order of deamination (leucine to KIC), hydroxylation, decarboxylation, and reduction (Route 3). If glucose is used as the sole substrate, Route 3 consists only of the hydroxylation, decarboxylation, and reduction steps, while Route 1 seems to contain a futile cycle between transamination and deamination from a pathway stoichiometric point of view (Fig. 3c). Surprisingly, Route 1 is more effective than Route 3 as demonstrated by the significantly higher IPDO titer when fed with the same concentration of precursors, i.e., leucine for Route 1 and KIC for Route 3. The reason behind this seems to be due to the lower catalytic capability of MFL on KIC than on leucine (this was experimentally confirmed, see results later), since the latter is regarded as the natural substrate of MFL. As a result, the metabolic flux channeling into Route 1 is expected to be much higher than that into Route 3 under physiological conditions. This implies that effective hydroxylation is the key step in the proposed pathway of IPDO production.

It seems that the higher metabolic flux of Route 1 is reflected by its more favorable thermodynamic profile compared with that of

Route 3 (Fig. 4). As depicted in Fig. 4a, b, Route 1 starts with a stronger exergonic reaction with a higher energy driving force than Route 3. The calculated thermodynamic profile of Route 1 with a gradual decrease of the ($\Delta_r G$) is more harmonized than that of Route 3. It would be interesting to further investigate how the three key constraints of a pathway, namely thermodynamics, enzyme kinetics, and biochemical properties of intermediates (e.g., permeability, toxicity, or stability), shape the reaction order and determine its overall activity. For the regulation and optimization of a reaction cascade, it would be also interesting to know if there exists an ideal thermodynamic profile as indicated by Fig. 4c. To this end, more intracellular and systematic information obtained under physiological conditions is needed. For both the working reaction orders, only one molecular NADPH is required for generating one molecular IPDO from leucine. In addition, they do not require expensive co-enzymes. Because the oxidative pathway of hydroxyl group formation does not lead to the loss of carbon atoms, the theoretical yield of diol is high. The theoretical yield of IPDO from glucose is estimated to be as high as 0.628 mol/mol glucose based on flux balance analysis (FBA) (Supplementary Figure 17). All these features indicate that it should be a robust and effective pathway.

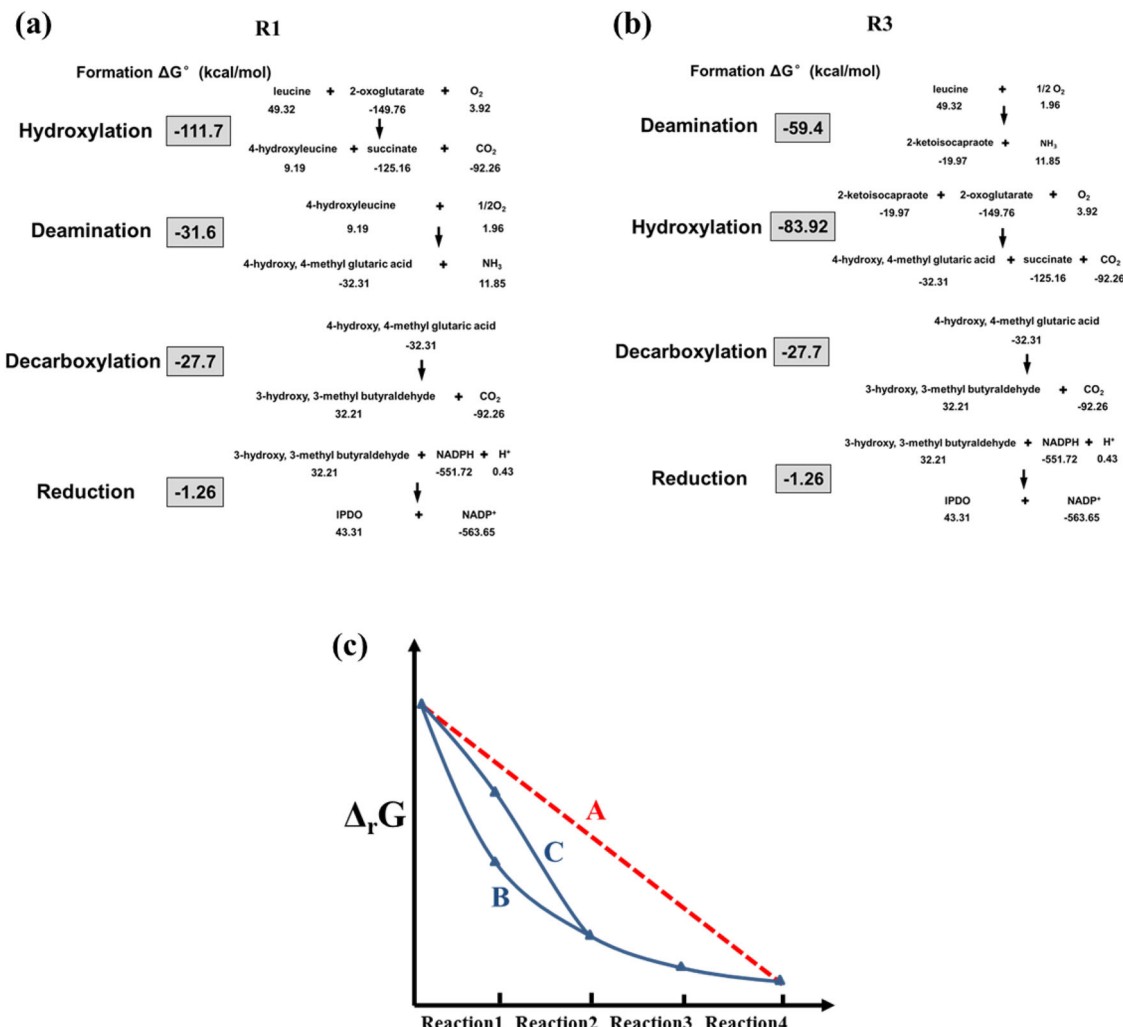

**Fig. 4 Thermodynamic profiles of reaction orders R1 and R3 for the biosynthesis of IPDO from leucine. a**, **b** Thermodynamic calculations of the reaction orders R1 and R3 of a four-step reaction cascade. The standard Gibbs free energy of formation of every non-natural product was estimated by using the group contribution method. **c** Schematic comparison of reaction driving forces ($\Delta_r G$) of three synthetic pathways that starts from the same substrate and end with the same product. We assumed that curve A (red dashed line) stands for a pathway with an ideal thermodynamic profile: the driving forces of all four reactions are identical. Curve B reflects the thermodynamic profile of Route 1 (**a**) that begins with a strong exergonic reaction with high energy driving force while curve C reflects the thermodynamic profile of Route 3 (**b**) that begins with a less thermodynamic favorable reaction.

**Optimization of the IPDO-producing route through enzyme mining and improved precursor supply**. To further optimize and demonstrate the efficiency of Route 1 for IPDO biosynthesis, we screened for hydroxylase and α-keto decarboxylase with improved catalytic efficiency. L-amino acid deaminase was not screened here, as AAD_vul from *P. vulgaris* showed the highest activity on leucine among all the reported L-amino acid deaminase from different species[31]. To this end, heterologous hydroxylases and α-keto decarboxylases of different origins were selected (Fig. 5a). All of the coding genes were synthesized with optimized codons. The resulting recombinant strains were inoculated and cultivated in FM-II medium with or without the addition of 25 mM leucine. The starting strain DL01 (named here as IP01 for enzyme screening purposes) harboring the plasmids pET-mfl and pZA-aky yielded 230.5 mg/L and 15.3 mg/L IPDO in the presence or absence of supplemented leucine (Fig. 5b). Two additional α-keto decarboxylases, PDC from *Z. mobilis* and THI3 from *Saccharomyces cerevisiae*, were selected to compare with KDC. The obtained strain IP02 harboring PDC accumulated 64.8 mg/L and 5.9 mg/L IPDO with or without additional leucine, which represented a reduction of 71.9% and 61.4% compared with those achieved in DL01. Similarly,

when KDC was replaced by THI3 in strain IP03, the titer of IPDO was significantly reduced to 21.4 mg/L and 1.4 mg/L, respectively (Fig. 5b). In addition to the hydroxylase MFL, four hydroxylases, namely AVI, IDO, GOX, and BPE from *Agrobacterium vitis*, *Bacillus thuringiensis*, *Gluconobacter oxydans*, and *Bordetella petrii*, respectively, were also selected for screening. Like MFL, they were all reported to stereospecifically catalyze the hydroxylation of amino acids at C-4 position[17]. Among the derived strains, IP04, IP05, and IP06 harboring AVI, IDO, and GOX, respectively, presented varying degrees of decreased production titers of IPDO compared with strain IP01. In contrast, IP07 harboring BPE showed the accumulation of 220.3 mg/L and 13.7 mg/L IPDO, respectively, in the presence and absence of additional leucine, indicating a high catalytic activity of BPE comparable to that of MFL in IPDO production. Taking together the screening results, the strain IP01 and IP07 turned out to be the two best IPDO-producing strains. Given that the $K_m$ value of MFL on leucine is lower than that of BPE[17], the strain IP01 was selected as the best strain for the follow-up optimization study.

In our designed pathway, leucine is one of the most important precursors for IPDO production. The results shown above

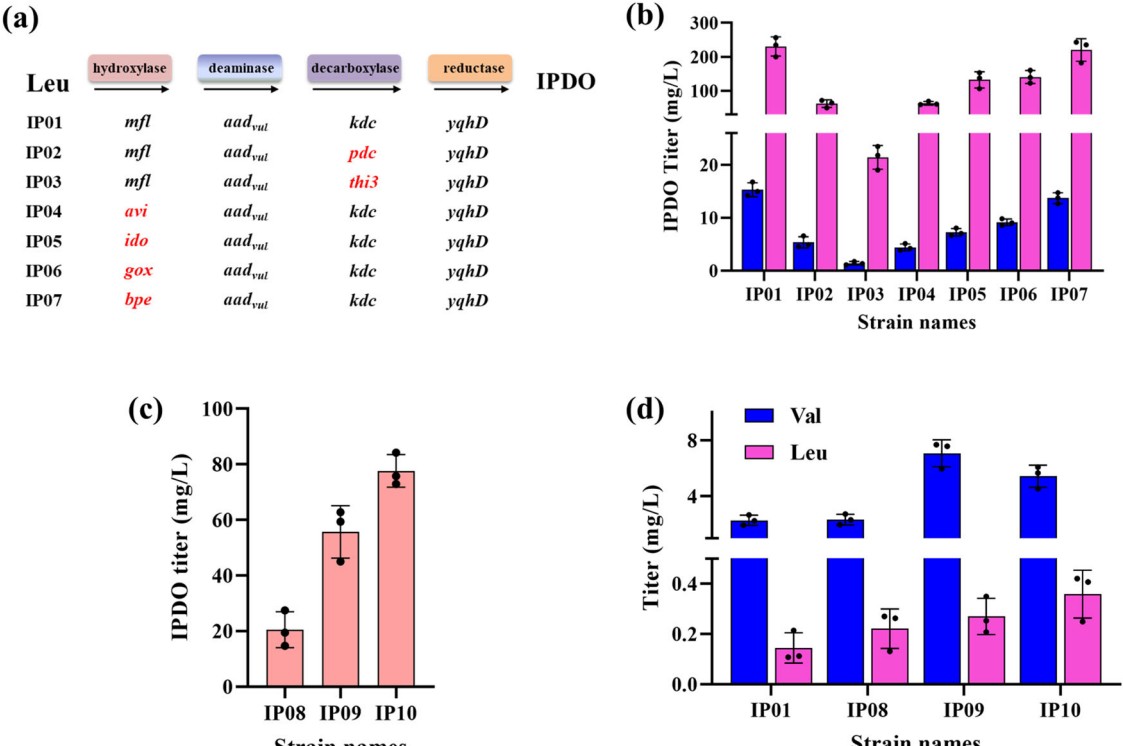

**Fig. 5 Optimization of the biosynthetic pathway to improve IPDO production in *E. coli*. a** IPDO production strain construction by screening hydroxylases and α-keto decarboxylase in the designed pathway from leucine to IPDO. **b** IPDO production of the constructed strains cultivated in the FM-II medium with (pink) and without (blue) the addition of leucine. **c** Increase IPDO production by the construction of strain IP08 overexpressing leucine operon (*leuA^{fbr}BCD* to enhance leucine supply and further strain IP09 overexpressing acetolactate synthase *alsS* and *ilvCD* to enhance 2-ketoisocapraote (KIC) supply. **d** Reducing the formation of the byproduct valine by constructing strain IP10 overexpressing tyrosine aminotransferase *tyrB*. IPDO and amino-acid titers were measured after 48 h of cultivation. Genes *ilvIH*, *ilvCD*, and *leuABCD* are from *E. coli*. Gene *alsS* are from *Bacillus subtilis*. For **b–d**, the average and s.d. of three biologically independent experiments are shown. Source data are provided as a Source Data file.

demonstrated that IPDO titer from the strain IP01 was significantly improved if additional leucine was supplemented into the medium containing glucose as substrate, implying that the native leucine supply was not sufficient for efficient production of IPDO from glucose. An enhancement of the leucine supply in IP01 is obviously necessary. For this purpose, leucine operon (*leuA^{fbr}BCD*[32]) from *E. coli* W3110 was overexpressed on the p15AS vector under the control of J23119 promoter (http://parts.igem.org/). The resulting strain IP08 showed an increase in the final IPDO titer from 15.3 mg/L (in IP01) to 20.5 mg/L without supplementing leucine (Fig. 5c). This was accompanied with a rise of the leucine concentration in the culture broth from 0.14 mg/L to 0.22 mg/L (Fig. 5d). KIC is the direct precursor of leucine. In a previous study, acetolactate synthase AlsS from *B. subtilis* was employed to increase the supply of cytoplasmic KIC[33]. In view of this, we further overexpressed *alsS* from *B. subtilis* as well as *ilvCD* genes from *E. coli* to yield strain IP09, thereby enhancing the flux from pyruvate to KIC. Compared with IP08, this strain showed a significant increase in IPDO titer, reaching 55.7 mg/L. However, 7.08 mg/L-valine was also accumulated in the culture broth of IP09, which was 3.04-fold higher than that of IP08 (Fig. 5d). Since KIC is the common precursor of leucine and valine, it is reasonable that overexpressing *alsS* and *ilvCD* led to a considerable part of the metabolic flux from KIC being shunted to the synthesis of valine. To address this issue, strain IP10 was constructed by overexpressing tyrosine aminotransferase TyrB from *E. coli* which promiscuously catalyzes the conversion of KIC to leucine but can't convert α-ketoisovalerate (KIV) to valine[34]. After 48 h of cultivation, a final titer of 77.6 mg/L IPDO was

achieved by the strain IP10, which was 1.62-fold higher than that of IP09. As expected, the leucine concentration in the broth further raised to 0.35 mg/L, whereas the valine decreased to 5.43 mg/L (Fig. 5d), indicating that overexpressing TyrB indeed improved the metabolic flux into the leucine synthesis pathway.

**Directed evolution of MFL for improved IPDO synthesis**. As already mentioned above, hydroxylation is the rate-limiting step in the proposed pathway of IPDO production. Previous research indicated that the catalytic activity of MFL on leucine is so poor that it is far from being suitable for technical application[17]. Therefore, to search for MFL mutants with higher hydroxylase activity EP-PCR was applied to construct a random MFL mutant library. For screening, we developed an approach based on the following principle: hydroxylation of leucine by MFL is coupled with oxidation of α-KG to succinate, so that the growth restoration of a succinate-auxotrophic strain is strictly dependent on the leucine hydroxylation, thus coupling cell growth to hydroxylase activity (Fig. 6a). A succinate-auxotrophic strain Δ3A (*E. coli* BL21 Δ*sucA*Δ*aceA*Δ*putA* (DE3)(pLysS))[35] was employed as a selection tool for screening positive mutants. The suitability of this strain for coupling growth rate with MFL activity was evaluated before the screening. Plasmid pET-mfl was transformed into the Δ3A strain and cultivated in M9 medium supplemented with 0.2 g/L leucine, 0.2 g/L α-KG, and 0.1 mM isopropyl β-d-1-thiogalactopyranoside (IPTG). As a control strain, Δ3A carrying pET28a was cultivated in parallel. Both strains grew poorly in M9 medium, although Δ3A-pET-mfl grew slightly better than the control strain after 60 h of cultivation (Supplementary Figure 18). This was mainly due to strains' poor growth environment and the heavy burden caused by

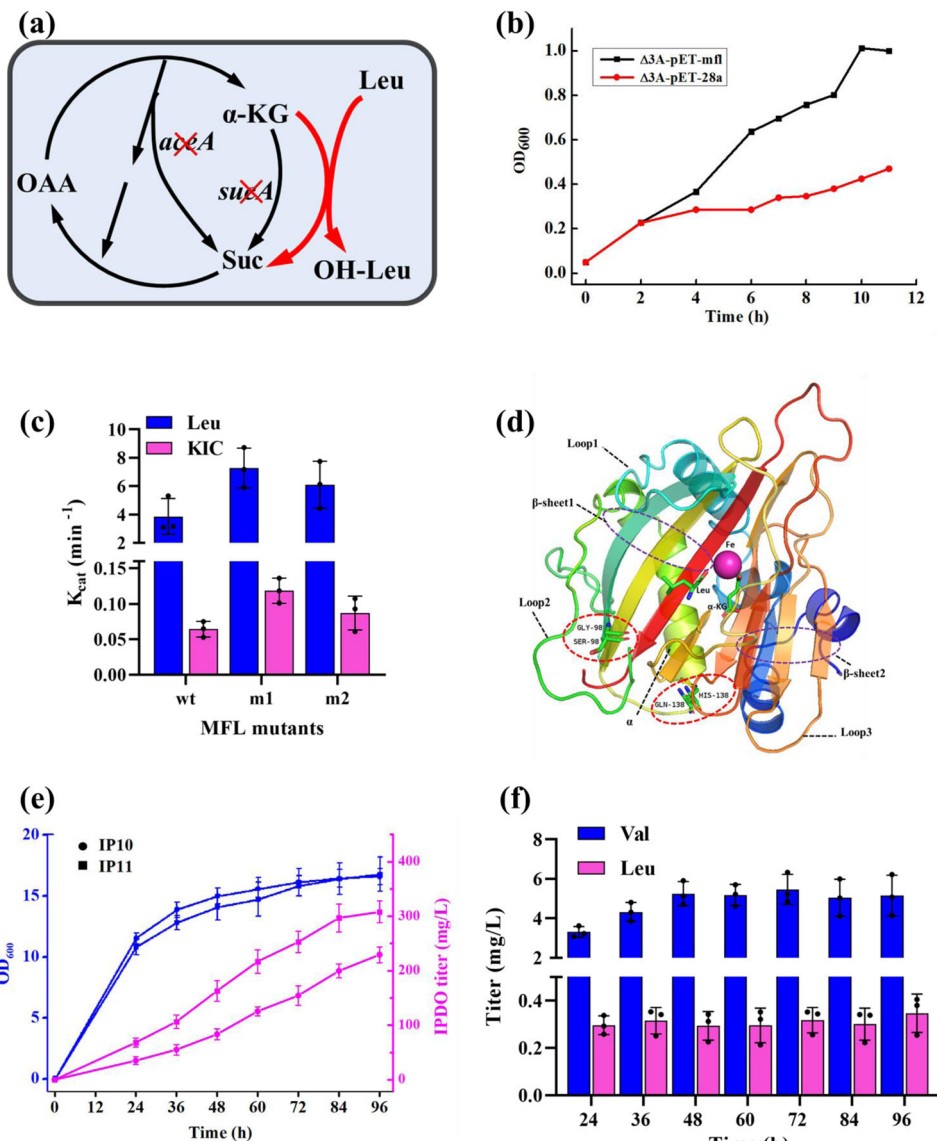

**Fig. 6 Directed evolution of MFL for screening MFL random mutants generated by EP-PCR. a** Schematic representation of the screening principle: hydroxylation of leucine by MFL is coupled with oxidation of α-KG to succinate (red arrows), and therefore the growth of a succinate-auxotrophic strain is strictly depended on leucine hydroxylation. **b** Growth curves of Δ3A strain carrying pET28a or pET-mfl in the M9 medium added with 4 g/L yeast. The initial OD was set as 0.05. **c** Enzymatic activities of MFL and its two mutants MFL$_{m1}$ and MFL$_{m2}$ towards leucine and KIC. **d** Positions of mutation residues found in MFL$_{m1}$ and MFL$_{m2}$. Fe$^{2+}$ is shown as a pink sphere. Leucine, α-KG, and two mutant residues are shown in sticks. L represents loop, αre presents alpha helix. **e** Growth curve (blue curves) and IPDO production (pink curves) in strain IP10 and IP11 cultivated in shake flasks. **f** Determination of valine and leucine concentrations in the fermentation broth of strain IP11 cultivated in shake flasks. Samples were taken every 12 h. For **c**, **e**, **f** the average, and s.d. of three biologically independent experiments are shown. Source data are provided as a Source Data file.

the addition of IPTG in the lag phase. To solve this problem, we enriched the medium and induced the expression of MFL when the cells were in the exponential phase. After adding 4 g/L yeast to the M9 medium both strains exhibited better growth; in particular, Δ3A-pET-mfl presented a notably faster growth after induction by IPTG (Fig. 6b). Hence, the modified M9 medium was used for the directed evolution of MFL.

An MFL mutant library with a capacity of 5000~6000 was constructed by EP-PCR. The quality of the mutant library was assessed before screening for mutants with higher activities. Sequencing results indicated that the quality of the mutant library was satisfactory and can be used for the subsequent screening of MFL mutants (Supplementary Figure 19). The MFL mutant library was then transformed into the Δ3A strain and seeded on M9 agar plates supplemented with 4 g/L yeast, 0.2 g/L leucine,

0.2 g/L α-KG, and 0.1 mM IPTG. After 48 h of cultivation, 18 large-sized colonies were selected and their hydroxylase activities were determined. The results indicated that two mutants MFL$_{m1}$ and MFL$_{m2}$ showed improved activities, reaching 1.9-fold and 1.5-fold of that of the wild type, respectively (Fig. 6c). The remaining mutants exhibited comparable or even decreased activities (Supplementary Figure 20). Surprisingly, in addition to improved hydroxylation of leucine, it was observed that promiscuous hydroxylation activities of MFL$_{m1}$ and MFL$_{m2}$ on KIC were also improved by 1.8-fold and 1.3-fold, respectively (Fig. 6c). Therefore, these two MFL mutants could promote IPDO production by simultaneously improving the performance of both synthetic routes (Route 1 and Route 3) (Fig. 3c) in the host strain.

Sequence alignment analysis showed that the residue His138 in the wild-type MFL was replaced by Gln in MFL$_{m1}$, whereas the

residue Ser98 was replaced by Gly in $MFL_{m2}$. As no crystal structure of MFL has been reported yet, homologous modeling was applied to predict the MFL structure. Homologous modeling generally requires that the sequence identity between the template protein and the target protein is higher than 30%[36]. After performing BLAST homology search against the NCBInr database, we found that MFL shares a 31.2% identity in amino acid sequence with the best hit MpDO, which is a member of the dioxygenases family (PF10014) (Supplementary Figure 21) and was chosen as the template for homologous modeling of MFL. It is observed that two β-sheets are packed against each other, forming a cup-shaped β-sandwich with a topology characteristic of the double-stranded β-helix fold, a classic structure that is shared by the family members. Residues involved in binding $Fe^{2+}$ and α-KG are strictly conserved between MFL and MpDO (Supplementary Figure 21), suggesting that they may share the same enzymatic mechanism. The substrate-binding site of MFL is formed by sheet 1 and three variable loops (Fig. 6d). It is suggested that the plasticity of the active site is very important for the catalysis of cupin dioxygenases[37]. The flexible configuration of the three loops leads to differences in the size and accessibility of the active site[38]. However, it should be noted that the modeled structures of the three loops may not be reliable due to the low sequence identity between MFL and the template MpDO. According to the modeling, position 138 is located on an α-helix at the opposite side of the active site pocket (Fig. 6d, Supplementary Figure 21). The S98G mutation is located in a loop region (Fig. 6d, Supplementary Figure 21). This loop was reported to play a role in adjusting the plasticity of the active site in MpDO through conformation changes[37]. Whether the S98G mutation has an effect on the conformation change of this loop needs further experimental verification.

To examine the performance of $MFL_{m1}$ on IPDO biosynthesis, $MFL_{wt}$ in strain IP10 was replaced by $MFL_{m1}$, yielding strain IP11. Batch fermentation with IP11 was performed in a shake flask. As shown in Fig. 6e, after 24 h of cultivation its OD reached 10.8 and gradually increased to 16.8 at 96 h, which is similar to that of the control strain IP10. IPDO production of IP11 was steadily increased in an almost linear way throughout the whole cultivation period and appeared to level off near the end of the cultivation, reaching a final titer of 0.31 g/L. It is remarkable that the IPDO titers from the strain IP11 were 1.3~2.0 times higher than that from strain IP10 in the cultivation time from 24 h to 96 h of cultivation, which is well correlated with the elevated MFL activity in strain IP11. This again confirms that the MFL-catalyzed step is the primary limiting step in the constructed IPDO synthetic pathway, and highlights the effectiveness of boosting MFL activity for further improvement of IPDO production. However, a high level of extracellular valine was still detected in the fermentation broth of strain IP11. The concentrations of leucine and valine in the culture broth of strain IP11 were maintained at levels comparable to what was observed in the strain IP10 during the whole cultivation time (Fig. 6f).

## Discussion

In this study, we put forward a general strategy to produce structurally diverse diols by extending amino acid catabolism in *E. coli*. As proof of concept, we synthesized ten structurally different diols to illustrate the versatility and feasibility of the designed platform pathway. Eight of the synthesized diols are branched-chain diols. This work thus enriched the repertoire of diols produced through biological routes.

In contrast to the predominantly existing synthetic pathways in which both hydroxyl groups were obtained through the reductive pathway, we came up with a platform pathway in which one of the hydroxyl groups in the diol product is introduced through an oxidative pathway. Indeed, both reductive and oxidative reactions need to overcome thermodynamic barriers. The reduction of carboxyl to carbonyl group is energetically constrained when only NAD(P)H is used as electron donor, and the oxidation of hydrocarbon to hydroxyl is also unachievable using $NAD(P)^+$ as the only electron acceptor without additional energy input. In fact, organisms have evolved enzymes that are able to counteract these thermodynamic barriers. One example of carboxyl reduction to carbonyl is found in gluconeogenesis, in which 3-phospho-D-glycerate is reduced to D-glyceraldehyde-3-phosphate (Supplementary Figure 22a). This conversion is fulfilled by two reaction steps that are separately catalyzed by phosphoglycerate kinase and glyceraldehyde-3-phosphate dehydrogenase so that in addition to NADH, and ATP is used to provide additional energy to circumvent the reduction potential barrier. For the oxidative pathway, one example of alkyl oxidation is the conversion of succinate to malate in the tricarboxylic acid (TCA) cycle (Supplementary Figure 22b). This conversion employs ubiquinone, which possesses a higher reduction potential than $NAD^+$, as an electron acceptor instead of $NAD^+$. These two examples represent two strategies that organisms adapt to power thermodynamically unfavorable reactions, i.e., either by coupling with energetic reactions or by investing an extra ATP. On one hand, ATP can be directly involved in the reaction to provide energy to overcome the thermodynamic barriers. On the other hand, when an indirect reaction coupling is employed, the energy released by ATP hydrolysis is used to generate the chemical bond with higher energy, and the cleavage of which can subsequently energize the unfavorable reaction. However, one significant advantage of the oxidative reaction for hydroxyl group formation in our route is that it provides a large driving force to the whole pathway and greatly reduces the difficulty of engineering strain for NADH/NADPH supply, thus making it more appealing compared with previously reported diol pathways. It is also important to note that the oxidative pathway is a more orthogonal one compared with the reductive one since ATP and NADH/NADPH are involved in so many other metabolic pathways. This is also an important feature from the perspectives of synthetic biology and systems metabolic engineering to achieve a highly effective synthetic metabolic pathway.

In fact, our proposed approach of combining an oxidative pathway with a reductive pathway is traceable in the biosynthetic pathways of some natural compounds. Hydroxytyrosol, an important aromatic triol that is extensively used as a potent antioxidant scavenger of free radicals[39], is generated through combining oxidative and reductive pathways for hydroxyl group formation[40,41]. The structure of this natural pathway bears some similarity to our proposed route, but the catalytic mechanism of the pathway enzymes and the reaction orders are very different. One key reaction in this route is the oxidation of tyrosine to dihydroxyphenylalanine, which is performed by tyrosine hydroxylase. This aromatic hydroxylase uses either tetrahydrobiopterin (BH4) or tetrahydromonapterin (MH4) as a coenzyme and shows promiscuous activity towards tyrosol or tyramine[42]. Moreover, a cofactor regeneration system is also required for the supply of the coenzyme. In contrast, the BCAA hydroxylase employed in our pathway fulfills its function with the only requirement on the more accessible ferrous ion $Fe^{2+}$ as the cofactor. In addition, in the case of IPDO production, MFL enabled hydroxylation of leucine or its corresponding precursor KIC, which is strikingly different from that of tyrosine hydroxylase.

As shown with the synthetic IPDO pathway, the reaction order is of significance for the performance of a synthetic pathway. The thermodynamic analysis seems to provide a useful guideline to

evaluate the possibilities of different reaction orders of a given pathway with the same substrate and product (Fig. 4a, b). A harmonized thermodynamic profile of $\Delta_r G$ values with a gradual decrease of the energetic driving forces for the four reactions in the IPDO pathway follow the order of hydroxylation>deamination>decarboxylation>reduction, and it turned out also to be the most effective reaction order in terms of productivity (kinetics). In terms of kinetics, it is interesting to note that both hydroxylase and deaminase exhibit activities towards leucine, especially the hydroxylase shows activity on leucine even at a physiologically relevant concentration $(K_m < 0.3\,\text{mM})$[17], while decarboxylase and reductase were proved to have no activities on leucine (Supplementary Figure 16). Furthermore, we noticed that α-ketoisocaproate (KIC) as intermediate is toxic. This metabolic constraint determines that the intracellular KIC should be maintained at a low level. Therefore, the subsequent reaction, either transamination of KIC to leucine in route 1 (R1) or hydroxylation of KIC to 4-hydroxy-KIC in route 3 (R3) should be both kinetically and thermodynamically favorable so as to proceed far from equilibrium. However, the enzymatic activity of the hydroxylase MFL on KIC is quite low (Fig. 5c), making R3 less competitive than route R1. Thus, our results highlight the coherence between thermodynamics and kinetics, an aspect that is often neglected in evaluating an artificial pathway.

We selected IPDO, an important branched-chain 1,n-alkanediol that is commercially attractive in the cosmetic industry, as an example to demonstrate how to improve the performance of the proposed diol biosynthetic route in *E. coli* through metabolic engineering. Strain engineering was mainly focused on improving leucine supply and hydroxylase activity. In strain IP10 we improved leucine supply by overexpressing tyrosine aminotransferase TyrB, which exhibits catalytic specificity for leucine synthesis from KIC but not for valine synthesis from KIV, resulting in thereby 1.6-fold higher IPDO titer. Despite this, accumulation of valine was still observed. To overcome this limitation in the future, a dynamic regulation system may be employed to dynamically control the expression abundance of the branched-chain amino acid aminotransferase IlvE that catalyzes the final step in BCAAs biosynthesis. Through the introduction of a dynamic genetic element, a high expression level of IlvE may be maintained in the exponential growth phase to ensure good cell growth, and then turned down to a lower level when cells reach the stationary phase to avoid shunting metabolic flux towards valine synthesis.

Another engineering target is hydroxylase. Hydroxylases are ubiquitous in nature. The potential physiological functions of hydroxylases include helping cells to compete with other bacterial species[17], and breaking down toxic compounds in the environment[43]. Owing to their functions in nature hydroxylases always exhibit a weak or conditionally-induced activity. Therefore, in this study, we employed a succinate-auxotrophic strain as a selection tool for the directed evolution of the hydroxylase MFL. After random mutation and screening, a mutant MFL (MFL$_{m1}$) was evolved which showed an 88% improved catalytic capability compared with MFL$_{wt}$. We are aware that single mutations may not be sufficient. For example, two previous studies aiming at improving the catalytic efficiency of BCAAs hydroxylase have obtained mutants that showed 4.7-fold to 6.1-fold higher activities with the introduction of two to three mutation sites, respectively. Interestingly, mutation sites of the screened mutants are separately found in the binding pocket and in the loop that controls the entrance of the substrate in both studies[44,45], indicating a synergistic contribution from conformational changes in the binding pocket and the loop. Therefore, it seems that introducing two or more mutation sites is more likely to construct mutants with increased catalytic activity. To verify this, we

constructed a mutation library in which 2~3 mutation sites were observed in most of the MFL mutants (Supplementary Figure 23). This MFL library was then transformed into Δ3A strain and 10 colonies with large sizes were picked up and their hydroxylase activities were measured. The results show that MFL activities of five of the ten selected mutants were improved by over 100%, the best of which (MFL$_{Hm1}$) reached 2.5-fold of that of the wild type (Supplementary Figure 24). Sequencing results revealed that there are two mutation sites in MFL$_{Hm1}$ (D57V/K235E), indicating that MFL variants with higher catalytic activity can be obtained through the combinational effect of two or more mutation sites. However, we want to point out that the two libraries constructed in this work are still not adequate to obtain substantial improvements in the catalytic activity of MFL. As we have got several hotspots of MFL from EP-PCR results, a state-of-the-art of method such as the CASTing method[46] should be adopted in future work to design a more focused library near these hotspots with multiple mutations. It is also highly desired to determine the crystal structure of MFL for further rational design or semi-rational design.

For the production of long-chain diols, industrial strains are generally cultivated under anaerobic or microaerobic conditions, as the diol products are highly reductive. In our proposed pathway oxygen is involved in the hydroxylation reaction. Therefore, we need to evaluate the overall oxygen requirement in further study. Especially, we will examine if microaerobic conditions can be applied, as $O_2$ is only involved in the hydroxylation step of the four enzyme cascades and the total $O_2$ requirement is expected not as high as required by aerobic bioprocesses where $O_2$ is mainly consumed for the respiration or the oxidation of the excess reducing power, as in the cases of the bioproductions of 2,3-butanediol, 1,3-butanediol, and 1,4-butanediol. It should also be mentioned that aerobic bioprocesses do not necessarily mean problems. In fact, aerobic bioprocesses have normally much higher biomass formation and correspondingly higher product concentration and productivity than anaerobic processes. The formation of by-products can be better controlled in aerobic processes than in anaerobic processes.

A series of enzymes that catalyze the hydroxylation of BCAAs at different carbon atom positions have been reported. We compared and classified these hydroxylases according to their catalytic sites. The rationales behind the choice of certain hydroxylases in our study are explained in the following. (1) At least three delta-specific (C5) branched-chained amino acid hydroxylases have been reported[24,43,47,48], namely GriE from Streptomyces DSM 40835, AVIDO from *Anabaena variabilis,* and IDO from *Bacillus thuringiensis*. GriE was selected in this study for several reasons: first, compared with the other two hydroxylases, GriE possesses a broader substrate spectrum. Except L-valine and L-isoleucine which do not have a C-5 position, GriE shows catalytic activities to various branched-chain amino acids and their derivatives, including L-leucine, norleucine, norvaline, L-allo-isoleucine, γ-methyl-L-leucine, and 4-hydroxy-leucine[24]. In contrast, AVIDO can merely catalyze the hydroxylation of L-leucine and norleucine[43], whereas IDO only catalyzes the hydroxylation of norleucine[47]. Second, the catalytic capabilities of GriE on various substrates have been characterized, which can be used to evaluate the fermentation results of the corresponding diol products. As mentioned in the manuscript, when the strain DL04 overexpressing GriE, KDC, AAD$_{vul}$, and YqhD was employed to produce different kinds of diols by using glucose and corresponding amino acid as co-substrates, the titers of the diols (2-M-1,4-BDO>>1,4-PTD>1,4-BDO) were in line with the catalytic efficiency of GriE towards the corresponding BCAAs (leucine>>norleucine>norvaline)[24]. Last but not least, the crystal structure of GriE has been determined in complex with the

substrate leucine and α-KG[48]. This not only facilitates us to get a better understanding of the catalytic mechanism of GriE, but also offers guidelines for the further engineering of GriE to improve its catalytic performance for the diol production. (2) Many hydroxylases that selectively catalyze C-4 hydroxylation of BCAAs have been reported. In this study, a total of five hydroxylases (MFL, AVI, IDO, GOX, and BPE) were alternatively employed to optimize the IPDO synthetic pathway. All of them were discovered by Sergey V. Smirnov et al.[17] through BLAST analysis. The kinetic parameters (including $K_m$ and $K_{cat}$ towards leucine) of these hydroxylases have been determined. Our results indicated that the catalytic efficiencies ($K_{cat}/K_m$ value) of these hydroxylases do not directly correlate with the IPDO titers. For instance, the strain IP06 that harbors GOX ($K_{cat}/K_m = 66 \pm 9$ mM$^{-1}$ min$^{-1}$) produced less IPDO compared with the strain IP01 that harbors MFL ($K_{cat}/K_m > 39$ mM$^{-1}$ min$^{-1}$) and the strain IP07 that harbors BPE ($K_{cat}/K_m = 10 \pm 4$ mM$^{-1}$ min$^{-1}$). This is mainly because the kinetic parameters of these enzymes were determined in vitro, but their performances in vivo are obviously influenced by additional factors, such as their expression levels and the reaction environments in vivo. In vivo studies involving C-4 hydroxylase are scarce. Although MFL is chosen as the best C-4 hydroxylase for IPDO production, its activity is still too low for industrial application. Therefore, it is important to us to develop C-4 hydroxylases with significantly improved activity by protein engineering in future work. (3) Differently from other BCAAs, isoleucine possesses two C-4 positions (C4$^1$-OH and C4$^2$-OH). In addition to HilA and HilB from Pantoea ananatis AJ13355[15] that selectively catalyze the hydroxylation of C4$^1$ and C4$^2$ atoms in isoleucine, L-Ile dioxygenase (IDO) from Bacillus thuringiensis is reported to be able to catalyze the hydroxylation of L-Ile at C4$^2$ atom[18]. But the catalytic capability of IDO is unknown. In the following study, we will apply IDO in the general pathway to produce 2-M-1,3-BDO, in comparison with HilB. (4) So far it is known that AVI from Agrobacterium vitis and BPE from Bordetella petrii are the only two hydroxylases that have activities on threonine. Similarly, in the following study, we will explore other threonine hydroxylases that exhibit considerable catalytic activity for 1,3-PDO synthesis.

We used norvaline as the precursor for the production of 1,3-BDO and 1,4-BDO, and norleucine for 1,3-PTD and 1,4-PTD. Although norvaline and norleucine are not natural amino acids, several efforts have been made to engineer E. coli strain to produce norvaline and norleucine from glucose[49,50]. In their studies, norvaline and norleucine can be accumulated by blocking the branched-chain synthetic pathway and enhancing the so-called keto acid chain elongation pathway. Under optimized conditions, up to 5 g/L norleucine were secreted into the culture broth[49]. Thus, the corresponding diols can be principally synthesized through the proposed general pathway by using glucose as the sole carbon source.

The majority of the diols verified in this study were synthesized by extending BCAA metabolisms. Principally, more diols and their derivatives can be generated based on this platform pathway by utilizing other amino acids as precursors. In addition to diols, triols can also be synthesized by cascading two hydroxylases in the platform. This form of hydroxylation can be found in nature[15,51]. For instance, since hydroxylases MFL and GriE catalyze the hydroxylation of BCAAs at C-4 and C-5, respectively, 1,2,4-butanetriol (used for the production of 1,2,4-butanetriol trinitrate, an important propellant) could be synthesized from norvaline by employing these two hydroxylases. In a similar manner, 1,3,4-pentanetriol and 2,3,4-trihydroxybutyric acid as nutritional additives can be obtained from norleucine and glutamate, respectively.

The titers of the diols in our study are currently quite low, but they can be further increased by applying more sophisticated

systems metabolic engineering approaches as demonstrated by the development history of industrial processes for the production of 1,3-propanediol and 1,4-butanediol. In view of the need and trend to transfer the fossil-based manufacturing of chemicals to a one based on renewable resources and the fast development in the biological sciences and engineering (especially in systems and synthetic biology and in bioprocess engineering)[52], we believe that the production of more bio-based diols will become industrially competitive, especially for those with optically active stereoisomers, biological synthesis with its high stereoselectivity has decisive advantages.

Overall, a platform pathway for the biosynthesis of structurally diverse diols, in particular branched-chain 1,n-diols, is elaborated in this study by harnessing the combinatory power and advantages of oxidative and reductive formation of hydroxyl groups. It expands the amino acid metabolism for manufacturing compounds of industrial significance.

## Methods

**Bacterial strains, plasmids, and cultivation.** The strains and plasmids used in this study are listed in Supplementary Data 1. E. coli strains Top10 and BL21(DE3) were used for gene cloning and expression. E. coli BL21 ΔsucAΔaceAΔputA (DE3) (pLysS) strain (named in this work as Δ3A) is a kind gift from the group of Professor Bruno Bühler[35] and was used for directed evolution of the hydroxylase MFL from Methylobacillus flagellatus. Genes encoding L-amino acid deaminase aad_vul [https://www.ncbi.nlm.nih.gov/protein/BAA90864.1/] and hydroxylases griE [https://www.ncbi.nlm.nih.gov/nuccore/KP211414.1/], ido [https://www.ncbi.nlm.nih.gov/protein/ADJ94127.1/], avi [https://www.ncbi.nlm.nih.gov/protein/ABG82019.1/], bpe [https://www.ncbi.nlm.nih.gov/protein/WP_012248355.1/], mfl [https://www.ncbi.nlm.nih.gov/protein/YP_546733.1/?report=fasta], gox [https://www.ncbi.nlm.nih.gov/protein/WP_011253196.1?], hilA [https://www.ncbi.nlm.nih.gov/protein/BAK13116.1/] and hilB [https://www.ncbi.nlm.nih.gov/protein/BAK13117.1] were synthesized by Genscript (Netherlands). Genes encoding α-keto decarboxylases kdc [https://www.ncbi.nlm.nih.gov/protein/WP_046124870.1/], pdc [https://www.ncbi.nlm.nih.gov/protein/WP_011241152.1/] and thi3 [https://www.ncbi.nlm.nih.gov/protein/ONH78451.1] were previously synthesized by YJ Zhang and stocked in our laboratory[27]. The pET28a plasmid was used to express and purify proteins in E. coli BL21(DE3). Primers for plasmid construction were listed in Supplementary Data 2. Standards of IPDO, 1,3-PDO, 1,3-BDO, 1,4-BDO, 2-methy-1,3-propanediol (MPO), and 1,4-PTD were purchased from Sigma-Aldrich (Germany). Standards of 4-hydroxy-leucine (4OH-Leu), 1,3-PTD, 2-E-1,3-PDO, 2-M-1,3-BDO, and 2-M-1,4-BDO were purchased from AKos GmbH (Germany).

For shake flask fermentation, individual colonies were first inoculated in 5 mL LB medium and cultivated overnight before inoculating the culture in 10 mL modified FM-II medium comprising 30 g/L glucose, 0.5 g/L MgSO$_4$·7H$_2$O, 3 g/L KH$_2$PO$_4$, 12 g/L K$_2$HPO$_4$, 4 g/L (NH$_4$)$_2$SO$_4$, 1 g/L yeast extract, 2 g/L monosodium citrate, 1 mg/L thiamine, 100 µg/L biotin, 10 mL/L US$^{Fe}$ trace element solution[53], and 0.2 g/L FeSO$_4$·7H$_2$O. When OD$_{600}$ reached 0.4-0.6, 1 mM IPTG was added to induce the expression of hydroxylase. The cultivation was carried out at 37 °C and 200 rpm for 48 h. After separating cells by centrifugation supernatants were filtered through a filter membrane (0.22 µm) and stored at −20 °C for later analysis by gas chromatography (GC) or gas chromatography-mass spectrometry (GC-MS).

**Plasmids construction.** Taking pET-mfl as an example, the procedure of plasmid construction is as follows: generation of a linearized vector with 15 bp extension homologous at its ends was achieved by using primers pET-vec-LF and pET-vec-LR with pET28a as a template. The synthesized mfl fragment and linearized vector were ligated in an In-Fusion cloning reaction using the In-Fusion HD Cloning Kits (Clontech, Takara, Japan). The ligation product was transformed into E. coli Top10 and cultivated on LB agar plate containing 50 mg/L kanamycin (Kan) at 37 °C overnight. Positive colonies were verified by colony PCR and sequencing.

**MFL library construction.** Error-prone PCR (EP-PCR) was used for the construction of MFL library. The PCR program consisted of 1 min at 95 °C, followed by 35 cycles of 94 °C for 20 s, 55 °C for 20 s, and 72 °C for 1 min, and finished by 72 °C for 2 min. PCR components were used in the following concentrations: 5 µL 10× PCR Buffer, 0.2 mM dNTP Mix, 2 µM forward and reverse primers, 5–10 ng genome of E. coli MG1655 DNA, 0.1 mM MnCl$_2$, 3.0 mM MgCl$_2$, and 0.01 U/µL Taq polymerase. The PCR product was purified via gel electrophoresis using the Macherey-Nagel™ NucleoSpin™ Gel and PCR Clean-up Kit (Macherey-Nagel, Germany). Next, the purified DNA was cloned into pET28a vector using In-Fusion® HD Cloning Kit. The ligation product was transformed into E. coli Top10 by electroporation (2.5 kV, 25 µF, 200 Ω). The next day, cells were resuspended in

2 mL LB medium and the plasmid library was extracted using the Macherey–Nagel™ NucleoSpin Plasmid Kit (Macherey-Nagel, Germany).

**Protein expression and purification**. Selected colonies on the M9 plates were inoculated in LB medium and cultivated overnight at 37 °C and 200 rpm. Plasmids were extracted and transformed into *E. coli* BL21 (DE3) using electroporation (2.5 kV, 25 μF, 200 Ω). On the next day, colonies were incubated in LB medium at 37 °C and 200 rpm and used for inoculation of 50 mL fresh LB medium in 300 mL baffled Erlenmeyer flasks and cultivated at 37 °C and 200 rpm. When $OD_{600}$ reached 0.5, 1 mM IPTG was added to induce protein expression. After cultivation overnight, cells were centrifuged and the cell pellet was resuspended in 1 mL lysis buffer (500 mM NaCl, 20 mM imidazole, 20 mM $NaH_2PO_4$, pH 7.4). Lysis of the cells was achieved by using three rounds (3 × 45 sec, at a speed of 6.0 m/sec) of FastPrep homogenization (MP Biomedicals™, USA). After centrifugation, the target enzyme was purified from the cell extract via His-tag purification by using a His SpinTrap column (GE Healthcare, USA) with an elution buffer (300 mM NaCl, 250 mM imidazole, 50 mM $NaH_2PO_4$, pH 8). Finally, the enzyme buffer was changed to 100 mM HEPES buffer (pH 7.5) using an Amicon® Ultra 0.5 mL Centrifugal Filter (Merck, Germany).

**Enzymatic activity assays**. To measure the activity of MFL and its mutants, 10 μM of the hydroxylase was added to 5 mM leucine or 2-ketoisocaproate (KIC), 5 mM ascorbate, 0.2 g/L $FeSO_4 \cdot 7H_2O$, 5 mM α-KG (pH 7.0), and 100 mM HEPES (pH 7.0). The reaction was carried out at 30 °C for 1 h with vigorous shaking, then terminated by incubation at 95 °C for 10 min, and followed by centrifugation. The formation of succinate was determined using the succinic acid test kit (R-Biopharm, Darmstadt, Germany).

To measure the activity of the decarboxylation and reduction catalyzed, respectively, by the α-keto decarboxylases KDC and the aldehyde reductase YqhD, 50 mM sodium phosphate buffer, 1 mM $MgSO_4 \cdot 7H_2O$, 0.5 mM thiamine, 0.2 mM NADH disodium salt, 0.3 μM purified YqhD, and 0.1 μM purified KDC was added into the reaction system. The reaction was initiated by adding 5 mM of a corresponding amino acid as substrate. The reaction was incubated at 37 °C for 0.5 h and the NADH concentration was monitored by following the decrease in absorbance at 340 nm. To determine the activity of threonine deaminase on 4-hydroxy-threonine, 20 μM pyridoxal-5-phosphate hydrate, 50 mM potassium phosphate buffer (pH 7.5), 2 μg/mL purified enzyme, and the solution containing 4-hydroxy-threonine were mixed and cultivated at room temperature.

**Analysis of diols by GC and GC-MS**. For measurements of diols in the fermentation broth, filtered culture supernatants were mixed with acetonitrile (ACN) and a derivatization reagent (comprising 5 mg/mL phenylboronic acid dissolved in 2,2-dimethoxypropane) at a ratio of 1:2:1 (v/v) and incubated by shaking at 600 rpm at room temperature for 10 min.

GC analysis was performed on a phenomenex HP-5 capillary column (30 m × 0.25 mm × 0.25 μm) using a Varian 3900 GC instrument equipped with a flame ionization detector (FID). 1 μL of each sample was injected using a splitless/split injector (splitless for 1 min, then split at a ratio of 1:10) operating at 270 °C. Nitrogen was used as the carrier gas at a column flow rate of 1.5 mL/min. The column oven temperature program used was: initial temperature 100 °C (hold for 2 min), then ramp at 10 °C/min to 220 °C, and finally ramp at 20 °C/min to 260 °C (hold for 5 min). The FID detector was operated at 300 °C. Quantification of diols was realized using external standard calibration.

GC/MS analysis for confirming the formation of targeted diols was carried out on an Agilent 7890B gas chromatograph coupled to an Agilent MSD 5977 mass spectrometer. The GC column used was an Agilent DB-5ms UI capillary column (30 m × 250 μm × 0.25 μm). 0.2 μL of each sample were injected at a split ratio of 1:5 to the Gerstel CIS injector operating at an initial temperature of 60 °C (hold for 0.1 min), then ramping at a rate of 12 °C/s to 270 °C (hold for 1 min). Helium was used as the carrier gas at a constant pressure of 80 kPa. The column oven was held at an initial temperature of 60 °C (hold for 2 min), then ramps at 20 °C/min to 270 °C (hold for 12 min). The MS was operated in scan mode in the range of 2–500 amu, with a solvent delay of 4.5 min. The temperatures of the MS Source and MS Quad were set at 230 °C and 150 °C, respectively. The confirmation of diol formation (identification) was done through the comparison of the mass spectra with those acquired by using the purchased chemical standards, as well as through the confirmation of identical GC retention times.

**Analysis of amino acid in the fermentation broth by HPLC**. For measuring the concentrations of leucine, isoleucine, and valine in the culture broth, an Ultimate 3000 HPLC (Thermo Scientific) equipped with a fluorescence detector (FLD 3100, Thermo Scientific) and a Kinetex® C18 Column (100 × 4.6 mm, 2.6 μm, Phenomenex, USA) was used. Amino acids were derivatized using the AccQ-Fluor Reagent Kit (Waters, USA) according to the instruction of the manufacturer. The mobile phases used were Eluent A: 140 mM sodium acetate and 0.1% ACN in Milli-Q $H_2O$ (pH 4.95), Eluent B: 40% Milli-Q $H_2O$ and 60% ACN, and Eluent C: 5% Milli-Q $H_2O$ and 95% ACN. The elution gradient was set as follows: starting at 100% Eluent A, linear changing to 65% Eluent A + 35% Eluent B in 20 min, followed by washing with 100% Eluent C for 10 min, and then reconditioning the column for

5 min with 100% Eluent A. The flow rate was 1 mL/min, the column oven was kept at 45 °C, the sample injection volume was 10 μL, and the FLD 3100 detector was operated at an excitation wavelength of 250 nm and an emission wavelength of 395 nm.

**Flux balance analysis of IPDO biosynthesis in *E.coli***. The optimal IPDO synthetic pathway and flux distribution were evaluated by FBA using the genome-scale metabolic model iY75_1357 for *Escherichia coli* str. K-12 substr. W3110. We expanded it with the IPDO pathway by adding the following four reactions using the software COPASI (http://copasi.org/)

(1) Leucine + 2-oxoglutarate + $O_2$→hydroxyleucine + succinate + $CO_2$
(2) 4-hydroxyleucine + 1/2 $O_2$→4-hydroxy,4-methylglutaric acid + $NH_3$
(3) 4-hydroxy,4-methylglutaric acid→3-hydroxy,3-methylbutyraldehyde + $CO_2$
(4) 3-hydroxy,3-methylbutyraldehyde + $NADPH+H^+$→IPDO+$NADP^+$

The resulting model was named iY75_1357_IPDO and applied for a constraint-based FBA, in which the glucose uptake rate was set at 15 mmol/g/DCW and the source of oxygen, nitrogen, phosphate, and sulfur were not constrained.

**Protein homology modeling**. The three-dimensional (3D) homology model of MFL was constructed by using Modeller 9.25[54]. Crystal structure of the dioxygenase MpDO from *Methylobium petroleophilum* PM1 (PDB code 3PL0, https://www.rcsb.org/structure/3PL0) at 1.91 Å resolution was selected as the template. Sequence alignment between the template and MFL was performed using the command "align2d" and homology modeling of MFL was generated by using the command "automodel". The number of constructed models was set to be 5. The models were then evaluated based on the values of Modeller objective function and DOPE assessment scores. The model with the highest DOPE score was selected for subsequent docking with leucine.

**Docking MFL with leucine**. Since $Fe^{2+}$ and α-KG were not included in the crystal structure of MpDO, they were docked into the conserved location of MFL after superpositioning with FTO (PDB code 3LFM, https://www.rcsb.org/structure/3LFM), another member of the cupin superfamily. Docking MFL with leucine was performed using the software Autodock_4.2.6, according to the published protocol[55]. The generated results were visualized and analyzed by using PyMOL_1.8 (http://www.pymol.org).

**Statistical analysis**. Statistical analyses were performed with GraphPad Prism 8 software.

**Reporting summary**. Further information on research design is available in the Nature Research Reporting Summary linked to this article.

## Data availability
All the relevant data supporting the findings of this study are available within the Article and its Supplementary Information file. Source data are provided with this paper.

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

## Acknowledgements

We thank Benedikt Schober for supporting in the directed evolution of MFL, especially for the results presented in Fig. 6b. Y.L. is supported by a Ph.D. scholarship from the Chinese Scholarship Council (CSC), which is gratefully acknowledged.

## Author contributions

A.Z., Y.L., and W.W. conceived the concept of the work. Y.L. designed and performed the experimental studies. Y.L. and W.W. did the analytic work. Y.L., W.W., and A.Z. were involved in data analysis and discussion and in writing the manuscript. A.Z. provided financial support and supervised the work.

## Funding

## Competing interests

A patent application (no. LU101726) based on this work has been filed. This patent covers the verifications of IPDO, 1,3-PDO, 1,3-BDO, 1,4-BDO,1,4-PTD, MPO, 2-M1,4-BDO, 2-M1,3-BDO, 2-E-1,3-PDO, and 1,3-PTD productions from different BCAAs via the proposed general diol biosynthetic route. Yongfei Liu and An-Ping Zeng are listed as inventors of the patent. Other author claims no competing interest.
