## [Peer Review File · Nature Communications]

Biosynthesis of structurally diverse diols via a general route combining oxidative and reductive formations of OH-groupsReviewers' Comments:

Reviewer #1:

Remarks to the Author:

Liu and colleagues embedded a pathway from amino acids to diols into natural *Escherichia coli* metabolism for the biological production of C3-C5 diols from renewable feedstocks. To this end, the synthetic four enzyme cascade employing amino acid hydroxylase, L-amino acid deaminase, α -keto acid decarboxylase and aldehyde reductase was developed. As is the case for the biotechnological production of branched-chain higher alcohols (10.1038/nature06450), one carbon dioxide is lost in the 2-oxo acid decarboxylase reaction and similar enzymes are employed (KDC from *Lactobacillus lactis*). KDC, L-amino acid deaminase AADvul from *Proteus vulgaris*, and aldehyde reductase YqhD from *E. coli* were used in combination with hydroxylase MFL from *Methylobacillus flagellatus* (for 2-methyl-1,3-propanediol, 3-methyl-1,3-butanediol (IPDO), 1,3-butanediol and 1,3-pentanediol), *Pantoea ananatis* hydroxylases hilA (for 2-ethyl-1,3-propanediol) and hilB (for 2-methyl-1,3-butanediol) or GriE from *Streptomyces* DSM 40835 (for methyl-1,4-butanediol, 1,4-butanediol (1,4-BDO) and 1,4-pentanediol). What was the rationale to use exactly these? How many were considered and scored based on published information vs. those tested in experiments in this work?

What was the rationale to choose IPDO for further analysis? Reaction cosubstrates and the number of reactions between route 3 and 1 in figure 3 B differ (although in figure 4 calculations initiate with leucine in both cases). The transamination in route 1 requires an amino donor such as glutamate or alanine? Has the ATP/NADH cost of regeneration of glutamate/alanine been considered in the calculation?

After screening some variants for some enzymes in the pathway, leucine supply was improved following published strategies. Next, conversion of 2-oxoglutarate to succinate in the TCA cycle was blocked to drive flux via the hydroxylase reaction as published before. The resulting strain required a large amount of yeast extract to grow and faster growing mutants were selected from a MFL hydroxylase mutant library constructed by error-prone PCR. Figure 6C should depict absolute values of kcat.

Can the knowledge gained with IPDO be transferred to another example or is a new set of optimization needed? A non-branched example should be chosen to tap on a different precursor pathway.

Reviewer #2:

Remarks to the Author:

This manuscript described a general approach for the production of branched-chain diols from amino acids by employing amino acid hydroxylase, L-amino acid deaminase, α -keto acid decarboxylase and aldehyde reductase. The authors demonstrated that the approach can be used for the production of ten diols, among which six are biologically synthesized for the first time. Especially, the authors showed that the pathway be used to produce 3-methyl-1,3-butanediol from glucose with a titer of 0.31 g/L. Generally, the research is interesting and the proposed pathway is new and may be used to produce some useful diols. However, there are some major issues that should be addressed.

1. Although the authors demonstrated the feasibility to produce different diols by adding corresponding amino acids as precursor, the titers of most diols are extremely low. Moreover, there are no metabolic pathways for the biological production of norvaline and norleucine. Thus, the authors should evaluate and discuss the feasibility of the pathway for producing specific diols since most of diols can be easily synthesized by chemical approaches and the price of these diols are much cheaper than the corresponding amino acid precursors.
2. For the production of diols especially long-chain diols (which are reduced products), anaerobic or microaerobic conditions are generally required. However, the proposed pathways require high aerobic condition (oxygen is needed for hydroxylase). How can this problem be solved? The authors should discuss this problem.
3. Section 2 of the results part is not very necessary and informative. The authors should focus on

engineering the pathway for the production of new diols.

4. Fig.2, It is not clear how threonine can be transferred into 1,3-PDO via the pathway. The authors should give a clear pathway in Fig.2. The authors mentioned that threonine is hydroxylated by a hydroxylase, and IlvA was used for the deamination of the product. Evidence should be given for the specific activity of IlvA. Also, the authors mentioned that PDC is inhibited by certain metabolites produced by the promiscuous catalysis of hydroxylase in DL05 and DL07. It is really doubtful. I suggest that the authors could carry out an in vitro test of the pathway for 1,3-PDO production by using purified enzymes.

5. What is the advantage for the production of 1,3-PDO via the proposed pathway. There are more than 5 new pathways which have been proposed and some pathways can give a titer higher than 7 g/L.

6. Enzyme assay in the supplementary material is not clearly described.

7. Most of branched diols have stereoisomers. How about the stereoisomeric compositions of the products?

8. Amino acid deaminase is also a limiting enzyme for the proposed pathway. Although the authors mentioned that L-amino acid deaminase from *Proteus mirabilis* is active for leucine, but whether it is also very active for hydroxy-leucine is unclear.

9. Fig.6, the fermentation profile of the control strain should also be provided.

10. Fig.6, Did the strain also accumulate other diols during the fermentation?

Reviewer #3:

Remarks to the Author:

In their manuscript, the authors identified MFL as the bottleneck enzyme, which they therefore engineered for increased catalytic activity toward leucine. In my comments, I will focus to the protein engineering part of this manuscript.

The authors state that "A previous research indicated that the catalytic activity of MFL on leucine is so poor that it is far from being suitable for technical application." Therefore, they applied a directed evolution strategy to identify variants with substantially increased catalytic activity. Upon screening a library of 6000 mutants, they identified two variants with a single mutation that had an increased catalytic activity. They then performed homology modelling and substrate docking to retrospectively interpret the functional role of the two exchanged amino acids.

Comments

1. The authors claim that they constructed "a random MFL mutant library" of 6000 variants. Ideally, this size is sufficient to cover most of the sequence space of single mutants. However, the authors did not assess the quality of the library: How was the coverage of sequence space? How many wild type sequences did the library contain? How many variants with more than one mutation?

2. The variant with the highest catalytic activity had less than a 2-fold activity as compared to the wildtype MFL. This increase in catalytic activity is much less than usually obtained by protein engineering, and much less than needed in order to make the MFL suitable for technical applications. I see two alternative explanations for their failure: 1. The mutant library was not random, but biased toward the wild type sequence or at least a small sequence space. 2. Single mutations are not sufficient to obtain an increased catalytic activity.

3. The authors used homology modelling to obtain a structure model of MFL, based on a template with 38% sequence similarity. At this low level of similarity, the methods they used (pairwise sequence alignment and structure modelling) is expected to provide a reliable core structure of the protein, but structurally flexible and sequence variable regions such as the substrate binding site ("plasticity of the active site is very important") cannot be reliably predicted.

4. The docking poses obtained by docking of a substrate to the binding site of an enzyme sensitively depend on the structure of the enzyme. Small structural variations in flexible regions are expected to result in relevant changes of the position and orientation of the substrate and have considerable effect to the docking score, which is used for ranking of the results. Despite these limitations, their docking

results can be used to identify amino acids that line the substrate binding site, but not to assign a functional role to individual amino acids.

5. The two single mutations that result in a slightly increased catalytic activity are found in the substrate binding site. Therefore, it is promising to focus to the substrate binding site for engineering for increased catalytic activity. There are well established methods such as the semi-rational CASTing method which are a successful strategy for the identification of variants with multiple mutations, all located at a site consisting of 10-20 positions. Based on their initial results, I suggest the authors to apply a state-of-the-art method to develop a library which contains variants with multiple mutations in the substrate binding site, in order to obtain variants with a substantially higher catalytic activity. An alternative approach would be to determine the crystal structure of the MFL-leucine complex and to do rational design of the substrate binding site.

REVIEWER COMMENTS

Reviewer #1 (Remarks to the Author):

Liu and colleagues embedded a pathway from amino acids to diols into natural *Escherichia coli* metabolism for the biological production of C3-C5 diols from renewable feedstocks. To this end, the synthetic four enzyme cascade employing amino acid hydroxylase, L-amino acid deaminase, α -keto acid decarboxylase and aldehyde reductase was developed. As is the case for the biotechnological production of branched-chain higher alcohols (10.1038/nature06450), one carbon dioxide is lost in the 2-oxo acid decarboxylase reaction and similar enzymes are employed (KDC from *Lactobacillus lactis*). KDC, L-amino acid deaminase AAD_{vul} from *Proteus vulgaris*, and aldehyde reductase YqhD from *E. coli* were used in combination with hydroxylase MFL from *Methylobacillus flagellatus* (for 2-methyl-1,3-propanediol, 3-methyl-1,3-butanediol (IPDO), 1,3-butanediol and 1,3-pentanediol), *Pantoea ananatis* hydroxylases hilA (for 2-ethyl-1,3-propanediol) and hilB (for 2-methyl-1,3-butanediol) or GriE from *Streptomyces* DSM 40835 (for methyl-1,4-butanediol, 1,4-butanediol (1,4-BDO) and 1,4-pentanediol).

1. What was the rationale to use exactly these (enzymes)? How many were considered and scored based on published information vs. those tested in experiments in this work?

Answer: A series of enzymes that catalyze the hydroxylation of branched-chain amino acids (BCAAs) at different carbon atom positions have been reported. We compared and classified these hydroxylases according to their catalytic sites. The rationales behind the choice of certain hydroxylases in our study are explained in the following.

(1) At least three delta-specific (C5) branched-chained amino acid hydroxylases have been reported¹⁻⁴, namely GriE from *Streptomyces* DSM 40835, AVIDO from *Anabaena variabilis* and IDO from *Bacillus thuringiensis*. GriE was selected in this study for several reasons: First, compared with the other two hydroxylases, GriE possesses a broader substrate spectrum. Except L-valine and L-isoleucine which do not have a C-5 position, GriE shows catalytic activities to various branched-chain amino acids and their derivatives, including L-leucine, norleucine, norvaline, L-allo-isoleucine, γ -methyl-L-leucine and 4-hydroxy-leucine et al³. In contrast, AVIDO can merely catalyze the hydroxylation of L-leucine and norleucine¹ while IDO only catalyzes the hydroxylation of norleucine². Second, the catalytic capabilities of GriE on various substrates have been characterized, which can be used to evaluate the fermentation results of the corresponding diol products. As mentioned in the manuscript, when the strain DL04 overexpressing GriE, KDC, AAD_{vul} and YqhD was employed to produce different kinds of diols by using glucose and corresponding amino acid as co-substrates, the titers of the diols (2M-1,4-BDO >> 1,4-PTD > 1,4-BDO) were in line with the catalytic efficiency of GriE towards the corresponding

BCAAs (leucine >> norleucine > norvaline)³. Last but not least, the crystal structure of GriE has been determined in complex with the substrate leucine and α -KG⁴. This not only facilitates us to get a better understanding of the catalytic mechanism of GriE, but also offers guideline for the further engineering of GriE to improve its catalytic performance for the diol production.

(2) Many hydroxylases that selectively catalyze C-4 hydroxylation of BCAAs have been reported. In this study, a total of five hydroxylases (MFL, AVI, IDO, GOX and BPE) were alternatively employed to optimize the IPDO synthetic pathway. All of these five enzymes were discovered by Sergey V. Smirnov et al. through BLAST analysis⁵. The kinetic parameters (including K_m and K_{cat} towards leucine) of these hydroxylases have been determined. Our results indicated that the catalytic efficiencies (K_{cat}/K_m value) of these hydroxylases don't directly correlate with the IPDO titers produced by the strains carrying them. For instance, the strain IP06 that harbors GOX ($K_{cat}/K_m = 66 \pm 9 \text{ mM}^{-1} \text{ min}^{-1}$) produced less IPDO compared with the strain IP01 that harbors MFL ($K_{cat}/K_m > 39 \text{ mM}^{-1} \text{ min}^{-1}$) and the strain IP07 that harbors BPE ($K_{cat}/K_m = 10 \pm 4 \text{ mM}^{-1} \text{ min}^{-1}$). This is mainly because that the kinetic parameters of these enzymes were determined *in vitro*, but their performances *in vivo* are obviously influenced by additional factors, such as their expression levels and the reaction environments *in vivo*. *In vivo* studies involving C-4 hydroxylase are scarce. Although MFL is chosen as the best C-4 hydroxylase for IPDO production, its activity is still too low for industrial application. Therefore, it is important to develop C-4 hydroxylases with significant improved activity.

(3) Differently from other BCAAs, isoleucine possesses two C-4 positions (C4¹-OH and C4²-OH). In addition to HilA and HilB from *Pantoea ananatis* AJ13355⁶ that selectively catalyze the hydroxylation of C4¹ and C4² atoms in isoleucine, L-Ile dioxygenase (IDO) from *Bacillus thuringiensis* is able to catalyze the hydroxylation of L-Ile at C4² atom⁷. The catalytic capability of IDO is unknown. In the following study, we will apply IDO in the general pathway to produce 2M-1,3-BDO, in comparison with HilB.

(4) So far it is known AVI from *Agrobacterium vitis* and BPE from *Bordetella petrii* are the only two hydroxylases that have activities on threonine. Similarly, in the following study, we will explore other threonine hydroxylases that exhibit considerable catalytic activity for 1,3-PDO synthesis.

2. What was the rationale to choose IPDO for further analysis? Reaction cosubstrates and the number of reactions between route 3 and 1 in figure 3 B differ (although in figure 4 calculations initiate with leucine in both cases).

Answer: We choose IPDO for further analysis based on the following two reasons. 1) IPDO is a commercially attractive compound, especially for its applications in cosmetic industry. We were approached by industry to develop a biological process for IPDO. 2) As mentioned in the manuscript title, one of the key novelties of our study is to explore a biosynthetic pathway for branch-chained diols. IPDO is a good

example of branch-chained diols, for which no natural metabolic pathway exists and no synthetic pathway has been reported to our knowledge.

Yes, Route 3 is a shortened metabolic pathway using α -ketoisocaproate as the key intermediate (“co-substrate”) instead of leucine as used in Route 1. In Figure 4 we calculated the thermodynamics of the two routes starting from the same “substrate” leucine for a better comparison. The deamination reaction of Route 3 (Figure 4b) is a fiction reaction which is now indicated by a dashed line and explained in the Figure legend.

3. The transamination in route 1 requires an amino donor such as glutamate or alanine? Has the ATP/NADH cost of regeneration of glutamate/alanine been considered in the calculation?

Answer: Yes, the amino donor is glutamate, and the cost of reducing power required is considered in the calculation. ATP is not need in this reaction. According to our flux balance analysis (FBA) result in Figure R1 (provided as Figure S6 in the supplementary material), glutamate maintains a balance by two degradation reactions (reaction 1 and 2) and one synthesis reaction (reaction 3). For reaction 1, the transamination reaction catalyzed by branched-chain-amino-acid aminotransferase converts α -ketoisocaproate to leucine. This reaction requires glutamate as the amino donor and costs one NADPH. In another degradation reaction (reaction 2), aspartate aminotransferase catalyzes the synthesis of aspartate and α -KG from glutamate and oxaloacetate. The synthesized α -KG, in turn, participates in the supply of glutamate through the catalysis of glutamate synthase (reaction 3), which utilizes either NADPH or NADH as reducing power. Besides, the production and consumption of the two reducing powers are balanced by transhydrogenases (reaction 4). The metabolic flux of this reaction is dependent on whether glutamate dehydrogenase is NADPH (a) or NADH-dependent (b).

Figure R1. FBA result of optimal metabolic flux distribution for IPDO production in *E. coli*.

4. After screening some variants for some enzymes in the pathway, leucine supply was improved following published strategies. Next, conversion of 2-oxoglutarate to succinate in the TCA cycle was blocked to drive flux via the hydroxylase reaction as published before. The resulting strain required a large amount of yeast extract to grow

and faster growing mutants were selected from a MFL hydroxylase mutant library constructed by error-prone PCR. Figure 6C should depict absolute values of k_{cat} .

Answer: Figure 6C is now modified according to the suggestion as follows to show k_{cat} value:

Figure R2. Revised figure 6C

5. Can the knowledge gained with IPDO be transferred to another example or is a new set of optimization needed? A non-branched example should be chosen to tap on a different precursor pathway.

Answer: The strategy applied in the optimization of IPDO production composed of three major parts: 1. mining and selecting pathway enzymes, 2. enhancement of precursor supply and 3. directed evolution of enzymes (here hydroxylase used as an example). All of them are “general” and, in principal, can be applied to other diol products. In our opinion, it is not essential to use a non-branched example for demonstrating pathway optimization. The main objective of this work is to demonstrate the principle and feasibility of diol biosynthesis by combined oxidative and reductive formations of hydroxyl groups, and we succeeded in demonstration the synthesis of 10 different diols. In addition, biosynthesis of non-branched diols has been well studied in literature^{8,9}.

Reviewer #2 (Remarks to the Author):

This manuscript described a general approach for the production of branched-chain diols from amino acids by employing amino acid hydroxylase, L-amino acid deaminase, α -keto acid decarboxylase and aldehyde reductase. The authors

demonstrated that the approach can be used for the production of ten diols, among which six are biologically synthesized for the first time. Especially, the authors showed that the pathway be used to produce 3-methyl-1,3-butanediol from glucose with a titer of 0.31 g/L. Generally, the research is interesting and the proposed pathway is new and may be used to produce some useful diols. However, there are some major issues that should be addressed.

1. Although the authors demonstrated the feasibility to produce different diols by adding corresponding amino acids as precursor, the titers of most diols are extremely low. Moreover, there are no metabolic pathways for the biological production of norvaline and norleucine. Thus, the authors should evaluate and discuss the feasibility of the pathway for producing specific diols since most of diols can be easily synthesized by chemical approaches and the price of these diols are much cheaper than the corresponding amino acid precursors.

Answer: It's true that the titers of the diols in our study are currently quite low. Because the main objective of this work is to demonstrate the principle and feasibility of diol biosynthesis by combined oxidative and reductive formations of hydroxyl groups (proof-of-concept). we haven't made much effort to improve the product titers except for IPDO. For the latter case we could improve the titer from 15.3 mg/L (IP01) to 310 mg/L (IP11) through metabolic engineering. We believe that the titer can be further increased by applying more sophisticated systems metabolic engineering approaches as demonstrated by the development of industrial processes for the production of 1,3-propanediol and 1,4-butanediol. For both of these processes the initial product titer was in the range of mg/L in the stage of proof-of-concept.

Although norvaline and norleucine are not natural amino acids, several efforts have been made to engineer *E. coli* strain to produce norvaline and norleucine from glucose^{10, 11}. In their studies, norvaline and norleucine can be accumulated through blocking the branched-chain synthetic pathway and enhancing the so-called "keto acid chain elongation pathway". Under optimized conditions, up to 5 g/L norleucine were secreted into the culture broth¹⁰. Thus, the corresponding diols can be principally synthesized through the proposed general pathway by using glucose as the solo carbon source.

Yes, the majority of C3-C5 diols (except for 1,3-propanediol) are presently mainly produced via chemical synthesis routes based on fossil resources, And the biological routes for most diols are yet economically far from competitive. In view of the need and trend to transfer the fossil-based manufacturing of chemicals to a one based on renewable resources and the fast development in the biological sciences and engineering (especially in systems and synthetic biology and in bioprocess engineering)¹², we believe that the production of more bio-based diols will become industrially competitive, especially for those with optically active stereoisomers, biological synthesis with its high selectivity has decisive advantages. In fact, we are evaluating such high-value and specific diols using the principal route proposed in this work.

2. For the production of diols especially long-chain diols (which are reduced

products), anaerobic or microaerobic conditions are generally required. However, the proposed pathways require high aerobic condition (oxygen is needed for hydroxylase). How can this problem be solved? The authors should discuss this problem.

Answer: We add the following contents in the discussion section: “For the production of long-chain diols, industrial strains are generally cultivated under anaerobic or microaerobic conditions, since the diol products are highly reductive. The proposed pathway requires aerobic conditions since oxygen is involved in the hydroxylation reaction. We will evaluate the overall oxygen requirement in further study. Especially we will examine if microaerobic conditions can be applied, as O₂ is only involved in the hydroxylation step of the four enzyme cascades and the total O₂ requirement is expected not as high as required by aerobic bioprocesses where O₂ is mainly consumed for the respiration or the oxidation of the excess reducing power, as in the cases of the bioproductions of 2,3-butanediol, 1,3-butanediol and 1,4-butanediol. It should also be mentioned that aerobic bioprocesses do not necessarily mean “problems”. In fact, aerobic bioprocesses have normally much higher biomass formation and correspondingly higher product concentration and productivity than anaerobic processes. The formation of byproducts can be better controlled in aerobic processes than in anaerobic processes”.

3. Section 2 of the results part is not very necessary and informative. The authors should focus on engineering the pathway for the production of new diols.

Answer: We believe that Section 2 is important for illustrating the reaction orders of the IPDO pathway. It helps us to get a better understanding of how the pathway (reaction cascade) performs *in vivo*. After identifying two possible routes that co-exist for the production of IPDO, we compare the metabolic fluxes of these two routes through thermodynamic analysis and kinetic analysis (Figures 3 and 4). Knowledge gained from this case study also provides some important guidelines for designing and evaluating different synthetic pathways that produce the same product from the same substrate. In addition, work in Section 2 also provides hints for further engineering the IPDO synthetic pathway. For instance, given that two possible routes co-exist for the IPDO production in the strain IP01, both leucine and 2-ketoisocaproate (KIC) should be taken into consideration as the substrates when rationally engineering the hydroxylase MFL to improve its activity.

4. Fig.2, It is not clear how threonine can be transferred into 1,3-PDO via the pathway. The authors should give a clear pathway in Fig.2. The authors mentioned that threonine is hydroxylated by a hydroxylase, and IlvA was used for the deamination of the product. Evidence should be given for the specific activity of IlvA. Also, the authors mentioned that PDC is inhibited by certain metabolites produced by the promiscuous catalysis of hydroxylase in DL05 and DL07. It is really doubtful. I suggest that the authors could carry out an *in vitro* test of the pathway for 1,3-PDO production by using purified enzymes.

Answer: In order to help readers to understand the proposed pathway more clearly, the reaction schemes of 1,3-PDO and the other nine diols from their corresponding

amino acids are now given in the supplementary material as Figure S1.

We carried out an in vitro enzymatic assay to verify the specific activity of IlvA on 4-hydroxy-threonine. Since 4-hydroxy-threonine is not commercially available, threonine hydroxylase BPE is used to synthesize 4-hydroxy-threonine from threonine. Both threonine hydroxylase BPE and threonine deaminase IlvA were separately ligated into pET-28a vector and transformed into *E. coli* BL21(DE3) strain. The two strains were induced by IPTG and cultivated at 30 °C overnight to overexpress BPE and IlvA, followed by purification by affinity chromatography on a His SpinTrap column (Figure R3). To verify the activity of IlvA on 4-hydroxy-threonine, we set one control group S0 and two experimental groups S1 and S2. The components of the three groups are shown in Table R1.

Figure R3. SDS-PAGE result of purified BPE and IlvA

Table R1. Components of the three reaction systems

	Components
S0	20 mM threonine + reaction buffer
S1	Adding 10 μ M BEP in S0, reaction for 16h
S2	Adding 10 μ M IlvA in S1, reaction for 8h

As can be seen from the results of HPLC analysis, when compared with the chromatograph of S0 (Figure R4) a new peak (retention time: 4.618 min) appears in

the chromatograph of S1 (Figure R5), accompanied with the decrease of threonine, which can only be 4-hydroxy-threonine, the product of threonine hydroxylation. After the addition of threonine deaminase IlvA, the area of this peak in S2 drops to about 38% (Figure R5, R6, R7), clearly indicating that IlvA has activity on 4-hydroxy-threonine. It is noteworthy that the amount of threonine in S2 drops to less than 2% (Figure R6, R7) which is mainly due to the fact that threonine is the natural substrate of IlvA. These results are provided as Figure S3 in the supplementary material.

Figure R4. Chromatograph of S0

Figure R5. Chromatograph result of S1

Figure R6. Chromatograph result of S2

Figure R7. Concentration changes of threonine and 4-hydroxy-threonine (in peak height) in S0, S1 and S2

The feasibility of the conversion of 4-hydroxyl- α -ketobutyric acid to 1,3-PDO catalyzed by KDC/PDC and YqhD in vitro has been demonstrated by several previous studies¹³⁻¹⁵. However, in our study the strain DL05 and DL07 harboring PDC can't produce 1,3-PDO from 4-hydroxyl- α -ketobutyric acid. Since one major difference between our DL05 and DL07 strains and the engineered strains in other studies is that threonine hydroxylase is employed in DL05 and DL07 in the 1,3-PDO synthetic route, so we believe that the promiscuous catalysis of threonine hydroxylase that inhibits the PDC activity can be a possible explanation for the failed production of 1,3-PDO by DL05 and DL07.

5. What is the advantage for the production of 1,3-PDO via the proposed pathway. There are more than 5 new pathways which have been proposed and some pathways

can give a titer higher than 7 g/L.

Answer: As mentioned in the manuscript, there are two general advantages for the diols produced through our proposed general pathway: (1) Compared to other diol synthetic pathways, less NAD(P)H is required for the formation of one molecular diol product. (2) Reactions in the proposed pathway are thermodynamically more favorable. To demonstrate this, we now carry out a thermodynamic analysis of all the five previously reported 1,3-PDO pathways as well as our proposed pathway (Figure R8).

Figure R8. Thermodynamic profiles of reactions in various 1,3-PDO synthetic route.

Note: Figure a, b and c are cited from the reference¹⁴

In the abovementioned pathways, pathway a is a combination of the conversion of glucose to glycerol with the natural 1,3-PDO pathway from glycerol. In other engineered metabolic pathways (b-e), the energy driving force of reactions catalyzed by NAD(P)H-dependent reductases are always quite low, needless to say that some reactions are thermodynamically unfavorable (such as the reactions catalyzed by homoserine dehydrogenase in route b, DHB dehydrogenase in route c and alcohol dehydrogenase in route e). On contrary, the hydroxylation, deamination and decarboxylation in route f (our proposal pathway) are thermodynamically more favorable reactions which can give the necessary push for the thermodynamically less favorable reductive formation of hydroxyl group in the last step. These results are provided as Figure S4 in the supplementary material.

6. Enzyme assay in the supplementary material is not clearly described.

Answer: We have now added a section “Enzymatic activity assays” in the “Materials and Methods” part of the manuscript to describe the assay methods used in this study, including that for hydroxylase (previously described in the section “4. Protein expression and purification”) and those for decarboxylase and reductase (previously given in the supplementary material).

7. Most of branched diols have stereoisomers. How about the stereoisomeric compositions of the products?

Answer: This is a good point. Although we haven’t conducted related studies on it, some hints can be found from the hydroxylase stereoselectivity. Theoretically speaking, the stereoisomeric compositions of the products are dependent on the stereoselectivity of the hydroxylase employed in the pathway. It’s reported that enzymes belonging to α -ketoglutarate-dependent dioxygenase superfamily possess high region- and stereo-selectivity¹⁶. As an important component, amino acid (and its homologs) hydroxylases catalyze the hydroxylation of substrate amino acid with high stereoselectivity¹⁷. Specifically, for some enzymes, such as GriE from *Streptomyces* DSM 40835³, HilAB from *Pantoea ananatis* AJ13355⁶ and IDO from *Bacillus thuringiensis*⁷, the stereoselectivity of their catalytic sites have been determined through H-NMR analysis; and they were found to catalyze hydroxylation of BCAAs with high stereoselectivity. Although the stereoselectivity of the other hydroxylases employed in this study (including AVI, BPE, MFL and GOX) is unclear so far, we speculate that they may also present high stereoselectivity. More detailed analysis should be done in this aspect. Unfortunately, standards for potential stereoisomers are hardly available.

8. Amino acid deaminase is also a limiting enzyme for the proposed pathway. Although the authors mentioned that L-amino acid deaminase from *Proteus mirabilis* is active for leucine, but whether it is also very active for hydroxy-leucine is unclear.

Answer: We agree. But the results shown in Figure 3 with the different combinations of reaction orders strongly indicate that the deaminase is also active for hydroxy-leucine, otherwise we would not reach 1,3-PDO formation in the reactions

routes R1 and R2.

9. Fig.6, the fermentation profile of the control strain should also be provided.

Answer: A revised figure 6e is added as suggested. The revised figure 6e is as follow:

Figure R9. Revised Figure 6e

We also revised the sentences in the manuscript to “As shown in Figure 6e, after 24h of cultivation its OD reached 10.8 and gradually increased to 16.8 at 96h, which is similar with that of the control strain IP10. IPDO production of IP11 was steadily increased in an almost linear way throughout the whole cultivation period and appeared to level off near the end of the cultivation, reaching a final titer of 0.31 g/L. It is remarkable that the IPDO titers from the strain IP11 were 1.34 ~ 1.95-times higher than that from strain IP10 from 24h to 96h of cultivation, which is well correlated with the elevated MFL activity in strain IP11.”

10. Fig.6, Did the strain also accumulate other diols during the fermentation?

Answer: For the measurements of IPDO in the cultures, samples collected from the fermentation broth were treated with phenylboronic acid (PBA) in 2,2-dimethoxypropane, so IPDO is quantified as cyclic phenylboronate (c-PB) ester by GC or GC/MS analysis. This method is also applied for the other 9 branched-chain diols in this study. Except for the IPDO's specific peak, no other noteworthy peaks were found in the GC chromatographs of the samples collected from strain IP11's fermentation broth, suggesting that no other diols were significantly accumulated during the fermentation.

Reviewer #3 (Remarks to the Author):

In their manuscript, the authors identified MFL as the bottleneck enzyme, which they therefore engineered for increased catalytic activity toward leucine. In my comments, I will focus to the protein engineering part of this manuscript.

The authors state that "A previous research indicated that the catalytic activity of MFL on leucine is so poor that it is far from being suitable for technical application." Therefore, they applied a directed evolution strategy to identify variants with substantially increased catalytic activity. Upon screening a library of 6000 mutants, they identified two variants with a single mutation that had an increased catalytic activity. They then performed homology modelling and substrate docking to retrospectively interpret the functional role of the two exchanged amino acids.

Comments

1. The authors claim that they constructed "a random MFL mutant library" of 6000 variants. Ideally, this size is sufficient to cover most of the sequence space of single mutants. However, the authors did not assess the quality of the library: How was the coverage of sequence space? How many wild type sequences did the library contain? How many variants with more than one mutation?

Answer: We added the following description in the supplementary material: "The quality of the MFL mutant library was assessed before screening for mutants with higher activities using the strain $\Delta 3A$. 10 colonies were randomly picked up from the library and sent for sequencing. Alignment of the mutants' sequences (Figure R10, provided as Figure S8 in the supplementary material) showed that nine in the ten mutants contained at least one site mutation. Among them, six contain one mutation site and three contain two mutation sites. The site mutations are distributed randomly throughout the MFL sequence, representing no bias toward any specific sequence space. This indicates that the quality of the mutant library was satisfactory and can be used for the subsequent high throughput screening of MFL mutants."

Figure R10. Distribution of mutation sites from 10 MFL mutants picked up from the library.

2. The variant with the highest catalytic activity had less than a 2-fold activity as compared to the wild type MFL. This increase in catalytic activity is much less than usually obtained by protein engineering, and much less than needed in order to make the MFL suitable for technical applications. I see two alternative explanations for their failure: 1. The mutant library was not random, but biased toward the wild type sequence or at least a small sequence space. 2. Single mutations are not sufficient to obtain an increased catalytic activity.

Answer: We think the first explanation should not be the dominating one in view of the sequencing results of the ten mutant strains randomly selected (see above our answer to comment 1). We agree that single mutations may not be sufficient to obtain an increased catalytic activity

Although the secondary substrate-binding sites have greater variation that may affect specificity and stereoselectivity of the hydroxylation reaction, BCAA hydroxylases share one common protein fold topology: a double-stranded β -helix fold (DSBH) as their core catalytic site and an HXD/EX_nH motif in the active site to coordinate the Fe(II) cofactor. Two previous studies aiming at improving the catalytic efficiency of BCAAs hydroxylase have obtained mutants that showed 4.7-fold to 6.1-fold higher activities with introduction of 2 to 3 mutation sites, respectively. Interestingly, mutation sites of the screened mutants are separately found in the binding pocket and in the loop that controls the entrance of the substrate in both studies^{18, 19}, indicating a synergistic contributions from conformational changes in the binding pocket and the loop. In another case, Zhang et al. applied a strategy similar to ours to screen L-isoleucine dioxygenase (IDO) mutants from a random library with around 9000 mutants²⁰. Their best mutant IDO^{M3} only exhibited a 1.5-fold higher activity compared with that of the wild type IDO. Overall, it seems that introducing two or more mutation sites are more likely to construct mutants with increased catalytic activity. In follow-up study, we will try to adjust the reaction system of the error-prone PCR to improve the mutation frequency of the library.

3. The authors used homology modelling to obtain a structure model of MFL, based on a template with 38% sequence similarity. At this low level of similarity, the methods they used (pairwise sequence alignment and structure modelling) is expected to provide a reliable core structure of the protein, but structurally flexible and sequence variable regions such as the substrate binding site ("plasticity of the active site is very important") cannot be reliably predicted.

Answer: Homologous modeling generally requires that the sequence identity between the template protein and the target protein is higher than 30%²¹. In addition to modeller, we also sought to use another online protein structure prediction server I-TASSER to model the structure of MFL using MpDO as the template protein as well. Alignment result demonstrates that except for some loop regions that are supposed to be flexible, the core structures of the two models are almost identical (Figure R11), especially in the substrate binding site, in which the two β -sheets pack against each other, forming a cup-shaped β -sandwich with a topology characteristic of the

double-stranded β -helix fold, a classic structure that is shared by the family members. Next, we mainly focused on the comparison of the substrate binding sites between MpDO and modeled MFL (Figure R12). Cupin superfamily members generally share a double-stranded β -helix fold containing two 4-stranded β -sheets, which correspond to β 8, β 9, β 13 and β 16 (sheet1), and β 11, β 12 and β 14 (sheet2) in MpDO (Figure R13). It is worth mentioning that the amino acid sequences of most of the β sheet between MpDO and MFL are conserved, implying accuracy of the double-stranded β -helix fold in the modeled MFL structure. Furthermore, several key residues involved in binding Fe^{2+} and α -KG and a $^{159}\text{PxPE}^{162}$ motif are strictly conserved between MpDO and its homologs²², and these residues are also found in MFL's sequence (blue and red frame in Figure R11), which demonstrated that MFL and MpDO shared the well-conserved active site features.

Figure R11. Alignment result of MFL structure modeled by Modeller and I-TASSER

4. The docking poses obtained by docking of a substrate to the binding site of an enzyme sensitively depend on the structure of the enzyme. Small structural variations in flexible regions are expected to result in relevant changes of the position and orientation of the substrate and have considerable effect to the docking score, which is used for ranking of the results. Despite these limitations, their docking results can be used to identify amino acids that line the substrate binding site, but not to assign a functional role to individual amino acids.

5. The two single mutations that result in a slightly increased catalytic activity are found in the substrate binding site. Therefore, it is promising to focus to the substrate binding site for engineering for increased catalytic activity. There are well established methods such as the semi-rational CASTing method which are a successful strategy for the identification of variants with multiple mutations, all located at a site consisting of 10-20 positions. Based on their initial results, I suggest the authors to apply a state-of-the-art method to develop a library which contains variants with multiple mutations in the substrate binding site, in order to obtain variants with a substantially higher catalytic activity. An alternative approach would be to determine the crystal structure of the MFL-leucine complex and to do rational design of the substrate binding site.

Answer to 4 and 5: we agree with these valuable comments. Yes, it is not useful to assign functional roles to individual amino acids located near the binding site only based on the docking result. Actually, we have already screened a good MFL mutant (Figure 6d) through a high throughput screening platform and the docking results of MFL and substrates are used to analyze and predict the mutation sites' potential effects on the interaction relationship between the enzyme and the substrate. This analysis provides a possible explanation of the mutation site's contribution to enzyme activity, especially for those that locate on the flexible regions (such as loop) which are difficult to rationally design.

In follow-up studies more sophisticated approaches such as CASTing can certainly be used to further increase the activity of the key enzymes like MFL.

References:

1. Correia Cordeiro, R.S., Enoki, J., Busch, F., Mugge, C. & Kourist, R. Cloning and characterization of a new delta-specific l-leucine dioxygenase from *Anabaena variabilis*. *J Biotechnol* **284**, 68-74 (2018).
2. Hibi, M. et al. Characterization of *Bacillus thuringiensis* L-isoleucine dioxygenase for production of useful amino acids. *Appl Environ Microbiol* **77**, 6926-6930 (2011).
3. Zwick, C.R., 3rd & Renata, H. Remote C-H Hydroxylation by an alpha-Ketoglutarate-Dependent Dioxygenase Enables Efficient Chemoenzymatic Synthesis of Manzacidin C and Proline Analogs. *Journal of the American Chemical Society* **140**, 1165-1169 (2018).
4. Lukat, P. et al. Biosynthesis of methyl-proline containing griselimycins, natural products with anti-tuberculosis activity. *Chem. Sci.* **8**, 7521-7527 (2017).
5. Smirnov, S.V. et al. A novel family of bacterial dioxygenases that catalyse the hydroxylation of

- free L-amino acids. *FEMS microbiology letters* **331**, 97-104 (2012).
6. Smirnov, S.V. et al. A novel l-isoleucine-4'-dioxygenase and l-isoleucine dihydroxylation cascade in *Pantoea ananatis*. *MicrobiologyOpen* **2**, 471-481 (2013).
 7. Ogawa, J. et al. A novel L-isoleucine metabolism in *Bacillus thuringiensis* generating (2S,3R,4S)-4-hydroxyisoleucine, a potential insulinotropic and anti-obesity amino acid. *Appl Microbiol Biotechnol* **89**, 1929-1938 (2011).
 8. Yim, H. et al. Metabolic engineering of *Escherichia coli* for direct production of 1,4-butanediol. *Nature chemical biology* **7**, 445-452 (2011).
 9. JWang J, L.C., Zou Y, Yan Y Bacterial synthesis of C3 C5 diols via extending amino acid catabolism. *Proc Natl Acad Sci U S A* **117**, 9 (2020).
 10. Anderhuber, N. et al. High-level biosynthesis of norleucine in *E. coli* for the economic labeling of proteins. *J Biotechnol* **235**, 100-111 (2016).
 11. Sycheva, E.V. et al. Overproduction of noncanonical amino acids by *Escherichia coli* cells. *Microbiology* **76**, 712-718 (2007).
 12. Lee, S.Y. & Kim, H.U. Systems strategies for developing industrial microbial strains. *Nature biotechnology* **33**, 1061-1072 (2015).
 13. Wang, C. et al. An Aldolase-Catalyzed New Metabolic Pathway for the Assimilation of Formaldehyde and Methanol To Synthesize 2-Keto-4-hydroxybutyrate and 1,3-Propanediol in *Escherichia coli*. *ACS synthetic biology* **8**, 2483-2493 (2019).
 14. Frazao, C.J.R. et al. Construction of a synthetic pathway for the production of 1,3-propanediol from glucose. *Scientific reports* **9**, 11576 (2019).
 15. Meng, H., Wang, C., Yuan, Q., Ren, J. & Zeng, A.P. An Aldolase-Based New Pathway for Bioconversion of Formaldehyde and Ethanol into 1,3-Propanediol in *Escherichia coli*. *ACS synthetic biology* **10**, 799-809 (2021).
 16. Gao, S.S., Naowarajna, N., Cheng, R., Liu, X. & Liu, P. Recent examples of alpha-ketoglutarate-dependent mononuclear non-haem iron enzymes in natural product biosyntheses. *Natural product reports* **35**, 792-837 (2018).
 17. Hibi, M. et al. Novel Enzyme Family Found in Filamentous Fungi Catalyzing trans-4-Hydroxylation of L-Pipecolic Acid. *Appl Environ Microbiol* **82**, 2070-2077 (2016).
 18. Sun, D. et al. Redesign and engineering of a dioxygenase targeting biocatalytic synthesis of 5-hydroxyl leucine. *Catalysis Science & Technology* **9**, 1825-1834 (2019).
 19. Sun, D. et al. Efficient Biosynthesis of High-Value Succinic Acid and 5-Hydroxyleucine Using a Multienzyme Cascade and Whole-Cell Catalysis. *Journal of agricultural and food chemistry* **67**, 12502-12510 (2019).
 20. Zhang, C. et al. A strategy for L-isoleucine dioxygenase screening and 4-hydroxyisoleucine production by resting cells. *Bioengineered* **9**, 72-79 (2018).
 21. Pearson, W.R. An introduction to sequence similarity ("homology") searching. *Current protocols in bioinformatics* **Chapter 3**, Unit3 1 (2013).
 22. Xu, Q. et al. Crystal structure of a member of a novel family of dioxygenases (PF10014) reveals a conserved cupin fold and active site. *Proteins* **82**, 164-170 (2014).

Reviewers' Comments:

Reviewer #1:

Remarks to the Author:

Authors responded well to the points raised by this reviewer. A transfer example would have been nice.

Reviewer #2:

Remarks to the Author:

The authors have addressed all of my questions. I have no other comments.

Reviewer #3:

Remarks to the Author:

Authors' response to reviewer's comment 1:

...10 colonies were randomly picked up from the library and sent for sequencing ... nine in the ten mutants contained at least one site mutation ... site mutations are distributed randomly throughout the MFL sequence ... This indicates that the quality of the mutant library was satisfactory...

Reviewer's comment:

For a protein of 251 aa, there are 5020 single amino acid mutants and 25 M double mutants. Even with an unbiased library of 6000 variants, the sequence space of single amino acid mutants is not covered, and the sequence space of double mutants is by far not covered. To assess the quality of their library, the authors analyzed only 10 variants. 1 variant was the wild type, in 3 variants the third nucleotide of the respective codon was exchanged. Therefore, I expect that between one third and one half of the variants code for the wild type amino acid sequence. Therefore, the quality of the mutant library is not satisfactory, because it is biased toward wild type.

Authors' response to reviewer's comment 2

...We think the first explanation should not be the dominating one in view of the sequencing results of the ten mutant strains randomly selected ... 4.7-fold to 6.1-fold higher activities with introduction of 2 to 3 mutation sites, respectively...

Reviewer's comment:

My previous comments are confirmed by the authors:

1. The library is biased toward wild type: 4 out of 10 had no exchange or an exchange in the third nucleotide (cf comment 1)

2. Three mutations are needed for 6-fold increase of activity.

As a consequence, the methods used by the authors and their protein engineering strategy which screens essentially a single mutant library (including many wild type enzymes) are not adequate to obtain substantial improvements of catalytic activity. Therefore, the authors should use a state-of-the-art method such as CASTing to substantially improve the catalytic activity of their bottleneck enzyme, as suggested previously.

Authors' response to reviewer's comment 3

...to model the structure of MFL using MpDO as the template protein as well. Alignment result demonstrates that except for some loop regions that are supposed to be flexible, the core structures of the two models are almost identical ...

Reviewer's comment:

In their response, the authors mix up "sequence similarity" and "sequence identity". In their manuscript, they state that the sequence similarity is 38%. In their response they cite that "Homologous modeling generally requires that the sequence identity between the template protein and the target protein is higher than 30% [21]". They are not aware that 38% sequence similarity corresponds to a sequence identity of 30% or even below.

In their response, they also claim that the high structural similarity of the core of the two model structures to each other and to the template supports the quality of their homology model. This is wrong, because the structural similarity is the consequence of the homology modeling process, which assumes that the core structures of proteins with a sequence identity higher than 30% are similar. Most importantly, the substrate binding site is not formed by the core β -sheet, but by variable loops, which have been modelled at a much lower quality. Thus, the modelled structure of the substrate binding site is not reliable.

Authors' response to reviewer's comment 4

...we agree with these valuable comments. Yes, it is not useful to assign functional roles to individual amino acids located near the binding site only based on the docking result...

Reviewer's comment:

The authors agree that the quality of the docking results is not sufficient to assign functional roles to individual amino acids, but experimental characterization of variants should be used instead. However, in their revised manuscript they still assign functional roles based on their docking results.

Authors' response to reviewer's comment 5

... the two single mutations are not found near the substrate binding site... knowledge obtained from the initial results can't provide us enough hints for determining the positions of the randomized amino acids for library construction. In addition, a high-resolution crystal structure of MFL is still not available which also hinders the selection of appropriate amino acids that locate near the binding pocket... we decided to further screen for better MFL mutants from a mutant library with a higher mutation frequency... the best of which (MFLHm1) reached 2.5-fold of that of the wild type...

Reviewer's comment:

In contrast to the claims of the authors, CASTing does not require a high-resolution structure of an enzyme-substrate complex. It is sufficient to use a good homology model with a docked substrate (as presented by the authors) in order to identify residues that contribute to the substrate binding site. This is a state-of-the-art method of designing a focused library of variants with multiple mutations.

Point-to-point reply (marked yellow) to the comments of Reviewers

Reviewer #1 (Remarks to the Author):

Authors responded well to the points raised by this reviewer. A transfer example would have been nice.

Answer: In follow-up study we plan to target more diol products with high industrial potential for further engineering.

Reviewer #2 (Remarks to the Author):

The authors have addressed all of my questions. I have no other comments.

Reviewer #3 (Remarks to the Author):

Authors' response to reviewer's comment 1:

...10 colonies were randomly picked up from the library and sent for sequencing ... nine in the ten mutants contained at least one site mutation ... site mutations are distributed randomly throughout the MFL sequence ... This indicates that the quality of the mutant library was satisfactory...

Reviewer's comment:

For a protein of 251 aa, there are 5020 single amino acid mutants and 25 M double mutants.

Even with an unbiased library of 6000 variants, the sequence space of single amino acid mutants is not covered, and the sequence space of double mutants is by far not covered.

To assess the quality of their library, the authors analyzed only 10 variants. 1 variant was the wild type, in 3 variants the third nucleotide of the respective codon was exchanged. Therefore, I expect that between one third and one half of the variants code for the wild type amino acid sequence. Therefore, the quality of the mutant library is not satisfactory, because it is biased toward wild type.

Answer: We admit that the library capacity in our study is by far not large enough to even cover all possible single and double amino acids mutation of MFL. Also, the mutation frequency of the constructed library is indeed quite low that the actual library capacity is further reduced due to the existence of a certain percentage representing the wild type amino acid sequence. Although the library capacity is not so large in our study, positive mutants with higher activities can still successfully be screened from it (even though they are still not so high as we would expect), which confirms the practicability of this screening approach in principle. Apart from the construction of a high quality library, the efficiency of the screening tool is also important to obtain better MFL mutants.

To further improve the MFL activity, we constructed a new MFL mutant library in which on average 2 ~ 3 mutation sites appeared in most of the mutants and applied the screening tool to selected better MFL mutants. As expected, the activity of the best of mutant reached 2.5-fold that of the wild type (Figure S13-14). The results of the two round selecting demonstrate that although the mutant library doesn't cover all possible mutants, a powerful tool can still help us to screen the best mutants from a myriad of candidates in the library.

Authors' response to reviewer's comment 2

...We think the first explanation should not be the dominating one in view of the sequencing results of the ten mutant strains randomly selected ... 4.7-fold to 6.1-fold higher activities with introduction of 2 to 3 mutation sites, respectively...

Reviewer's comment:

My previous comments are confirmed by the authors:

1. The library is biased toward wild type: 4 out of 10 had no exchange or an exchange in the third nucleotide (cf comment 1)
2. Three mutations are needed for 6-fold increase of activity.

As a consequence, the methods used by the authors and their protein engineering strategy which screens essentially a single mutant library (including many wild type enzymes) are not adequate to obtain substantial improvements of catalytic activity. Therefore, the authors should use a state-of-the-art method such as CASTing to substantially improve the catalytic activity of their bottleneck enzyme, as suggested previously.

Authors' response to reviewer's comment 5

... the two single mutations are not found near the substrate binding site... knowledge obtained from the initial results can't provide us enough hints for determining the positions of the randomized amino acids for library construction. In addition, a high-resolution crystal structure of MFL is still not available which also hinders the selection of appropriate amino acids that locate near the binding pocket... we decided to further screen for better MFL mutants from a mutant library with a higher mutation frequency... the best of which (MFLHm1) reached 2.5-fold of that of the wild type...

Reviewer's comment:

In contrast to the claims of the authors, CASTing does not require a high-resolution structure of an enzyme-substrate complex. It is sufficient to use a good homology model with a docked substrate (as presented by the authors) in order to identify residues that contribute to the substrate binding site. This is a state-of-the-art method of designing a focused library of variants with multiple mutations.

Answer to 2 and 5: As mentioned in the response to comment 1, we agree that the constructed libraries are biased toward wild type, and single or even double mutations on MFL are not adequate to reach 6-fold or higher increase of activity. As shown by the 2.5-fold activity increase of the mutant MFL_{Hm1}, in the revised manuscript we didn't obtain MFL mutants with satisfactory activity after two rounds of screening. More mutation sites are required to increase the possibility of generating mutants with higher activities

Since little is known about the crystal structure of MFL and its catalytic mechanism, we initially applied the EP-PCR method to determine the hotspots that have substantial effects on the improvement of its activity. In the next step we will use CASTing method to construct the new libraries by randomizing several amino acid residues that are near the hotspots and select better mutations from it. We believe it is more effective to increase the activity of MFL by combining EP-PCR and CASTing assay. Since we have got several hotspots of MFL from EP-PCR results,

we will pay great efforts on the evolution of MFL by applying the CASTing method in follow-up studies.

Again, we want to mention that the novelty of this study lies in providing a general platform route for the microbial production of structurally diverse diols and this novelty will not be damaged without the further improvement of MFL activity.

We therefore correspondingly revised the discussion part of the manuscript.

Authors' response to reviewer's comment 3

Reviewer's comment:

In their response, the authors mix up "sequence similarity" and "sequence identity". In their manuscript, they state that the sequence similarity is 38%. In their response they cite that "Homologous modeling generally requires that the sequence identity between the template protein and the target protein is higher than 30% [21]". They are not aware that 38% sequence similarity corresponds to a sequence identity of 30% or even below.

In their response, they also claim that the high structural similarity of the core of the two model structures to each other and to the template supports the quality of their homology model. This is wrong, because the structural similarity is the consequence of the homology modeling process, which assumes that the core structures of proteins with a sequence identity higher than 30% are similar. Most importantly, the substrate binding site is not formed by the core β -sheet, but by variable loops, which have been modelled at a much lower quality. Thus, the modelled structure of the substrate binding site is not reliable.

Answer: We apologize for the mistakes. MFL shares a 38.35% identity in amino acid sequence with MpDO. We revised the "sequence similarity" into "sequence identity" in the manuscript.

As stated by the reviewer, the substrate binding site of MFL is actually formed by the β -sheet 1 and three variable loops (revised Figure 6d), and the modeled structures of the three loops are not reliable due to the low sequence identity between MFL and the template MpDO. Therefore, we have now deleted the alignment results of MFL and MpDO, and MFL structures modeled by using Modeller and I-TASSER in the supplementary material. In addition, we have deleted the following sentences:

"In addition to modeller, we also sought to use another online protein structure prediction server I-TASSER to model the structure of MFL using MpDO as the template protein as well. which demonstrated that MFL and MpDO shared the well-conserved active site features."

and added the following sentences in the manuscript:

"After performing BLAST homology search against the NCBI database, we found that MFL shares a 38.35% identity in amino acid sequence with the best hit MpDO, which is a member of the dioxygenases family (PF10014) and was chosen as the template for homologous modeling of MFL. It is observed that two β -sheets are packed against each other, forming a cup-shaped β -sandwich with a topology characteristic of the double-stranded β -helix fold, a classic structure that is shared by the family members. Residues involved in binding Fe^{2+} and α -KG are strictly conserved between MFL and MpDO (Figure S10), suggesting that they may share the same enzymatic mechanism. The substrate binding site of MFL is formed by the sheet 1 and three variable loops (Figure 6d). It's suggested that the plasticity of the active site is very important for

the catalysis of cupin dioxygenases³⁷. The flexible configuration of the three loops can lead to differences in the size and accessibility of the active site³⁸. However, it should be noted that the modeled structures of the three loops may not be reliable due to the low sequence identity between MFL and the template MpDO.”

Authors' response to reviewer's comment 4

...we agree with these valuable comments. Yes, it is not useful to assign functional roles to individual amino acids located near the binding site only based on the docking result...

Reviewer's comment:

The authors agree that the quality of the docking results is not sufficient to assign functional roles to individual amino acids, but experimental characterization of variants should be used instead. However, in their revised manuscript they still assign functional roles based on their docking results.

Answer: We agree with this reviewer's comment, and deleted the sentences associated with assigning functional roles of the mutation sites on MFL_{m1}, MFL_{m2} and MFL_{Hm1} based on the docking results of MFL. Instead, we add the following sentence in the revised manuscript: “Whether the S98G mutation has an effect on conformation change of this loop need further experimental verification.”

Reviewers' Comments:

Reviewer #3:

Remarks to the Author:

The authors have addressed my comments and revised the manuscript accordingly.

Point-to-point reply (marked yellow) to the comments of Reviewers

Reviewer #3 (Remarks to the Author):

The authors have addressed my comments and revised the manuscript accordingly.

Answer: thank you for your positive comment.